# Shift orbifolds, decompactification limits, and lattices

**Dan Israël,**[a] **Ilarion V. Melnikov,**[b] **Yann Proto**[c]

[a]*Sorbonne Université, CNRS, Laboratoire de Physique Théorique et Hautes Énergies, LPTHE, F-75005 Paris, France*

[b]*Department of Physics and Astronomy, James Madison University, Harrisonburg, VA 22807, USA*

[c]*Bethe Center for Theoretical Physics, Universität Bonn, Bonn, Germany*

*E-mail:* israel@lpthe.jussieu.fr, melnikix@jmu.edu, yproto@uni-bonn.de

ABSTRACT: We describe the general shift orbifold of a Narain CFT and use this to investigate decompactification limits in the heterotic Narain moduli space. We also comment on higher rank theories and describe some applications to the CFT based on the Leech lattice and its shift orbifolds.

## 1 Introduction

The main purpose of this note is to clarify the consistency conditions in a large class of orbifold conformal field theories (CFTs). A familiar way to identify necessary conditions is by demanding that the partition function of the orbifold theory is modular invariant, but the arguments for modular invariance can be subtle, nor is it clear that they are sufficient: for instance, a putative partition function can have negative or non-integer coefficients in its expansion while being modular-invariant. A recent discussion of such issues can be found in [1].

We will consider a particularly simple class of orbifolds, where the parent theory is just the Narain heterotic CFT [2]. In such a theory the Hilbert space decomposes into sectors $\mathcal{H} = \oplus_{\boldsymbol{p}} \mathcal{H}_{\boldsymbol{p}}$ labeled by lattice points $\boldsymbol{p} \in \Gamma$, an even self-dual lattice of signature $d, d+16$, and we take a "shift orbifold," by which we mean that we gauge the action of a finite symmetry group $G$, and this action is block-diagonal with respect to the decomposition.

Typically we use orbifold CFTs to construct new theories with new spectra of operators and new global symmetries. General shift orbifolds of Narain CFT do not really provide us with new theories: if the orbifold is consistent, then it yields another Narain CFT, which we could also obtain by moving in the moduli space $\mathfrak{M}_{d,d+16}$ of the parent theory. Nevertheless, we think the results obtained here are instructive for several reasons. First, we hope that a complete solution in this case can be extended to more general asymmetric orbifolds of Narain CFTs [3] to recover and clarify the known consistency conditions; the perspective could potentially be useful in the more ambitious goal of classifying such orbifolds. Second, we expect that the lessons learned can also be applied to more general parent theories where the orbifold action involves a shift in a toroidal sector which is combined with another action in a different sector, which might be realized by a non-linear sigma model with a curved target space or a Gepner model.[1] Finally, even in the class of pure shift orbifolds there are intriguing questions with answers that illustrate some of the beautiful interplay between orbifold CFTs and lattices.

An example of such a question is just this: suppose we take a shift orbifold by a group $G$ of a Narain CFT with some specific choice of moduli $\rho \in \mathfrak{M}_{d,d+16}$. We know that the resulting theory corresponds to a CFT with moduli $\rho_G \in \mathfrak{M}_{d,d+16}$. What is $\rho_G$ in terms of $\rho$ and $G$? The question becomes particularly sharp when $d = 0$, so that we are really discussing shift orbifolds of ten-dimensional heterotic string theories. In this case there are two inequivalent lattices, $\Lambda_{16}$ and $\Lambda_8 + \Lambda_8$, and starting with one of them, say $\Lambda_{16}$, we would like to know which shift orbifolds lead to the other lattice. We will see that there is a simple criterion to distinguish such group actions, but it requires us to consider decompactification limits of shift orbifolds of the 9-dimensional heterotic string of the sort studied in [4]. As a by-product of formulating our criterion we also complete the discussion begun in [4] to show that the small radius limit of the 9-dimensional heterotic string is a decompactification if and only if the the Wilson line is rational.

We emphasize that we are discussing a classic subject, and although it is one that is not as well-known as it should be, we suspect that many (perhaps all ?) of our results are known to orbifold practitioners. For instance, various aspects of shift orbifolds were already discussed in the context of ten-dimensional non-supersymmetric string theories [5, 6].[2] A classic review [8] contains a useful discussion (see especially their appendix A.4). A lucid presentation of general shifts can be found in [9], as well as the PhD thesis [10].

While specific shifts have been used in countless orbifold constructions since then, we hope that our general discussion is at least illuminating even if not novel. In what follows, we will show that the "general shifting method" of [10] does describe the general shift orbifold construction, and we will then answer the question posed above by giving a criterion for establishing which shift orbifold of a ten-dimensional supersymmetric heterotic string yields an inequivalent theory. In the last section we will see that our criterion also provides a

---

[1]Understanding such quotients in the context of heterotic flux compactification was the original motivation for our work.

[2]This was generalized to other dimensions in a systematic fashion in [7].

perspective on higher rank lattices—in particular the Leech lattice and its relation to the other 23 Niemeier lattices.

## Acknowledgements

IVM's work was partially supported by the Humboldt Research Award and the Jean d'Alembert Program at the University of Paris–Saclay, as well as the Educational Leave program at James Madison University. The work of YP has received funding from the European Research Council (ERC) under the Horizon Europe (grant agreement No. 101078365) research and innovation program. We are grateful to A. Font for useful discussions and for suggesting the ten-dimensional shift orbifold question, and we thank H. Parra De Freitas for discussions, shared notes, and for pointing out a crucial reference; we thank both of them for comments on the manuscript. We also thank G. Chenevier for precious clarifications on Kneser neighbors, and G. Lockhart for valuable discussions.

## 2   Conventions and notation for Narain CFT

We begin by setting up conventions for the heterotic Narain compactification to $d$ dimensions.[3]

### 2.1   Lattice set up

Consider $\mathbb{R}^{d,d+16}$ equipped with the Minkowski metric $\eta$ given as above:

$$\eta = \begin{pmatrix} 0 & \mathbb{1}_d & 0 \\ \mathbb{1}_d & 0 & 0 \\ 0 & 0 & \mathbb{1}_{16} \end{pmatrix} . \tag{2.1}$$

We denote the corresponding inner product by $\cdot$, so that $v_1 \cdot v_2 = v_1^{\mathrm{t}} \eta v_2$ for any two vectors $v_{1,2} \in \mathbb{R}^{d,d+16}$. Next we choose a fiducial embedding of the lattice in the Lorentzian vector space

$$\Gamma = \Lambda_{d,d} + \Lambda_8 + \Lambda_8 \simeq \Lambda_{d,d} + \Lambda_{16} \subset \mathbb{R}^{d,d+16} , \tag{2.2}$$

so that the lattice inner product is given by $\cdot$. We denote the generators of $\Lambda_{d,d}$ by $\boldsymbol{e}_I$ and $\boldsymbol{e}^{*I}$, with $I = 1, \ldots, d$. These are null vectors satisfying $\boldsymbol{e}_I \cdot \boldsymbol{e}^{*J} = \delta_I^J$.

The $\Lambda_8$ and $\Lambda_{16}$ are even self-dual Euclidean lattices of rank 8 and 16 respectively. For each $\Lambda_8 \subset \mathbb{R}^8$ we choose the generators to be simple roots $\boldsymbol{\alpha}_i$ with inner product $\boldsymbol{\alpha}_i \cdot \boldsymbol{\alpha}_j$ normalized so that roots have length squared 2, and we denote the two mutually orthogonal sets of roots by $\boldsymbol{\alpha}_i$ and $\boldsymbol{\alpha}_i'$. We will sometimes abuse the notation and combine the simple roots into a single set $\boldsymbol{\alpha}_1, \ldots, \boldsymbol{\alpha}_{16}$, with the first 8 corresponding to the first $\Lambda_8$ factor, and

---

[3]This is a well-known story. A classic review is [11]. A recent overview can be found in introductory sections of [12], and our lattice conventions and notation mostly follow [13].

the last 8 to the second $\Lambda_8$ factor. If, instead, we wish to use the $\Lambda_{16}$ presentation, then we take the $\boldsymbol{\alpha}_1, \ldots, \boldsymbol{\alpha}_{16}$ to be a basis for

$$\Lambda_{16} = \mathrm{Span}_{\mathbb{Z}}\{\boldsymbol{\beta}_1,\ \boldsymbol{\beta}_2,\ \ldots,\ \boldsymbol{\beta}_{16},\ \boldsymbol{w}_+\}\ , \tag{2.3}$$

where the $\boldsymbol{\beta}_i$ are the simple roots of $D_{16} = \mathfrak{spin}(32)$, while $\boldsymbol{w}_+ = (\frac{1}{2}, \ldots, \frac{1}{2})$ is the highest weight of the positive chirality spinor representation.

Every lattice point $\boldsymbol{p} \in \Gamma$ is uniquely written as

$$\boldsymbol{p} = \mathrm{w}^I \boldsymbol{e}_I + \mathrm{n}_I \boldsymbol{e}^{*I} + \boldsymbol{L}\ , \tag{2.4}$$

where

$$\boldsymbol{L} = \sum_i \ell^i \boldsymbol{\alpha}_i\ , \tag{2.5}$$

and w, n, $\ell^i$ are integer coefficients. It will also be convenient for us to fix a Cartan–Killing basis for $\mathbb{R}^{16}$ with orthonormal basis vectors $v_\alpha$, $\alpha = 1, \ldots, 16$, with respect to which

$$\boldsymbol{L} = \sum_\alpha \ell^\alpha v_\alpha\ . \tag{2.6}$$

For any $\boldsymbol{p} \in \Gamma$

$$\boldsymbol{p} \cdot \boldsymbol{p} = 2\mathrm{n}_I \mathrm{w}^I + \boldsymbol{L} \cdot \boldsymbol{L} \in 2\mathbb{Z}\ . \tag{2.7}$$

Specializing to $d = 1$, we denote the heterotic moduli by a pair $(r, A)$, where $A \in \mathbb{R}^{16}$ is the set of Wilson line parameters. A choice of $(r, A)$ determines a point $\rho$ in the Grassmannian $\mathrm{Gr}(1, 17)$ through the orthogonal basis for $\mathbb{R}^{1,17}$ consisting of orthogonal vectors $\widetilde{\pi}$, $\pi$, and $\pi_\alpha^\circ$, $\alpha = 1, \ldots, 16$

$$\begin{aligned} \widetilde{\pi} &= \boldsymbol{e} - \left(r^2 + \tfrac{1}{2}A \cdot A\right)\boldsymbol{e}^* - A\ , \\ \pi &= \widetilde{\pi} + 2r^2 \boldsymbol{e}^*\ , \\ \pi_\alpha^\circ &= v_\alpha + (v_\alpha \cdot A)\boldsymbol{e}^* \end{aligned} \tag{2.8}$$

normalized to

$$\widetilde{\pi} \cdot \widetilde{\pi} = -2r^2\ , \qquad\qquad \pi \cdot \pi = 2r^2\ , \qquad\qquad \pi_\alpha^\circ \cdot \pi_\beta^\circ = \delta_{\alpha\beta}\ . \tag{2.9}$$

## 2.2 Vertex operators

The heterotic worldsheet theory consists of the Narain CFT with $d + 16$ left-moving chiral bosons $X_{\mathrm{L}}^I(z)$, $\mathcal{X}_{\mathrm{L}}^a(z)$, and $d$ right-moving chiral bosons $X_{\mathrm{R}}^I(\bar{z})$. For a geometric description we think of $X_{\mathrm{L}}^I$ and $X_{\mathrm{R}}^I$ as the holomorphic and antiholomorphic components of the compact bosons describing the torus, while the $\mathcal{X}_{\mathrm{L}}^a$ can be thought of as representing the heterotic

worldsheet (Weyl) fermions.[4]

At a generic point in the moduli space the Narain CFT has a $\mathfrak{u}(1)_{\mathrm{L}}^{\oplus d+16} \oplus \mathfrak{u}(1)_{\mathrm{R}}^{\oplus d}$ Kac-Moody symmetry, and the primary operators with respect to this symmetry are the vertex operators $\mathcal{V}_{\boldsymbol{p}}$ labeled by lattice points in $\Gamma$. These have weights $h_{\mathrm{L}}(\boldsymbol{p})$, $h_{\mathrm{R}}(\boldsymbol{p})$ that depend on the moduli. The operator's spin is determined by $\boldsymbol{p}$ alone:

$$s(\boldsymbol{p}) = h_{\mathrm{L}}(\boldsymbol{p}) - h_{\mathrm{R}}(\boldsymbol{p}) = \frac{\boldsymbol{p} \cdot \boldsymbol{p}}{2} = \mathrm{n}_I \mathrm{w}^I + \tfrac{1}{2} \boldsymbol{L} \cdot \boldsymbol{L} \ , \tag{2.10}$$

while the right-moving weight depends on $\boldsymbol{p}$ and the moduli, and for $d = 1$ is given by

$$h_{\mathrm{R}}(\boldsymbol{p}) = \frac{1}{4r^2} \left( \widetilde{\pi} \cdot \boldsymbol{p} \right)^2 = \frac{1}{4r^2} \left( \mathrm{n} - \left( r^2 + \tfrac{1}{2} A \cdot A \right) \mathrm{w} - A \cdot \boldsymbol{L} \right)^2 \ . \tag{2.11}$$

## 2.3 Symmetries of the Narain CFT

We will be interested in the action of a group $G$ on the vertex operators $\mathcal{V}_{\boldsymbol{p}}$ of the following form: for any element $g \in G$ the action is

$$g \circ \mathcal{V}_{\boldsymbol{p}} = U(g, \boldsymbol{p}) \mathcal{V}_{\varphi_g(\boldsymbol{p})} \ , \tag{2.12}$$

where $\varphi_g(\boldsymbol{p})$ is a lattice automorphism, and $U(g, \boldsymbol{p})$ is a phase. Consistency with group multiplication requires the phases and lattice isomorphisms to obey

$$U(g_2, \varphi_{g_1}(\boldsymbol{p})) U(g_1, \boldsymbol{p}) = U(g_2 g_1, \boldsymbol{p}) \ , \qquad \varphi_{g_2}(\varphi_{g_1}(\boldsymbol{p})) = \varphi_{g_2 g_1}(\boldsymbol{p}) \ . \tag{2.13}$$

This action should also be compatible with the OPE, which places further constraints on the phases $U(g, \boldsymbol{p})$: when $\varphi_g = \mathrm{id}$ for all $g \in G$ $U$ is a map $U : G \to \mathrm{Hom}(\Gamma, \mathrm{U}(1))$; more generally, the constraint on the $U(g, \boldsymbol{p})$ involves a choice of cocycle, but we will not need those details in what follows.

The action of $g$ on the Narain CFT is not in general a symmetry because it acts on the moduli. Writing the parameter dependence of the right-moving weight explicitly on $\rho \in \mathrm{Gr}(d, d + 16)$, for every $g \in G$ there is a map $\mu_g : \mathrm{Gr}(d, d + 16) \to \mathrm{Gr}(d, d + 16)$ defined by demanding that for all $\boldsymbol{p} \in \Gamma$

$$h_{\mathrm{R}}(\mu_g(\rho); \boldsymbol{p}) = \varphi_g^*(h_{\mathrm{R}}(\rho; \boldsymbol{p})) = h_{\mathrm{R}}(\rho; \varphi_g(\boldsymbol{p})) \ . \tag{2.14}$$

An action by $g$ is a symmetry of the Narain CFT with moduli $\rho$ if and only if $\mu_g(\rho) = \rho$.

## 3 General shift orbifolds

We now restrict our attention to $G$ a finite symmetry group of the Narain CFT whose action leaves invariant all of the Cartan Kac-Moody currents (and their superpartners), with $\varphi_g(\boldsymbol{p}) =$

---

[4]The full heterotic string contains, in addition, the right-moving fermion $\psi_{\mathrm{R}}^I(\bar{z})$—these are the superpartners of $X_{\mathrm{R}}^I$, as well as the Minkowski degrees of freedom for $\mathbb{R}^{1,9-d}$ and the $bc$–$\beta\gamma$ ghost system. These degrees of freedom (and the accompanying right-moving GSO projection) will not play a role in our discussion.

$\boldsymbol{p}$ for all $g \in G$ and all $\boldsymbol{p} \in \Gamma$, so that the action on the vertex operators is a pure phase:

$$g \circ \mathcal{V}_{\boldsymbol{p}} = U(g, \boldsymbol{p}) \mathcal{V}_{\boldsymbol{p}} . \tag{3.1}$$

From the preceding section, we see that the phase $U$ belongs to $\mathrm{Hom}(\Gamma, \mathrm{U}(1))$. Because $\Gamma$ is a self-dual lattice, any such homomorphism takes the form

$$U(g, \boldsymbol{p}) = e^{2\pi \mathrm{i} s_g \cdot \boldsymbol{p}} , \qquad\qquad s_g \in \mathbb{R}^{d, d+16} . \tag{3.2}$$

We will focus on the case where $G$ acts effectively, i.e. $U(g, \boldsymbol{p}) = 1$ for all $\boldsymbol{p} \in \Gamma$ if and only if $g = \mathrm{id}$. In this situation $G$ must be abelian, and we choose its generators so that

$$G = \mathbb{Z}_{k_1} \times \mathbb{Z}_{k_2} \times \cdots \mathbb{Z}_{k_N} , \tag{3.3}$$

where the $k_a$ are $N$ positive integers, with $k_1 > 1$ and $k_a$ dividing $k_{a+1}$. We then write the group elements in an additive notation $\ell = [\ell_1, \ell_2, \ldots, \ell_N]$, where $\ell_a \in \mathbb{Z}/k_a\mathbb{Z}$, and the action of $G$ is determined by $N$ shift vectors $s_a \in \Gamma \otimes_{\mathbb{Z}} \mathbb{Q}$:

$$U(\ell, \boldsymbol{p}) = \exp\{2\pi \mathrm{i} \sum_a \ell_a s_a \cdot \boldsymbol{p}\} . \tag{3.4}$$

Since $\Gamma$ is self-dual, and the group action requires that $k_a s_a \cdot \boldsymbol{p} \in \mathbb{Z}$ for all $a$, it follows that the action is determined by $N$ lattice elements $\boldsymbol{\sigma}_a = k_a s_a \in \Gamma$.

**A convenient basis**

Moreover, since $G$ acts effectively, we can choose the generators so that the $\boldsymbol{\sigma}_a$ are primitive and independent lattice elements that can be completed to a basis for $\Gamma$:

$$\Gamma = \mathrm{Span}_{\mathbb{Z}}\{\boldsymbol{\sigma}_1, \boldsymbol{\sigma}_2, \ldots, \boldsymbol{\sigma}_N, \boldsymbol{\xi}_{N+1}, \boldsymbol{\xi}_{N+2}, \ldots, \boldsymbol{\xi}_{2d+16}\} . \tag{3.5}$$

Self-duality of $\Gamma$ implies that there are dual elements $\boldsymbol{\sigma}^{*a}$ and $\boldsymbol{\xi}^{*i}$, such that

$$\Gamma = \mathrm{Span}_{\mathbb{Z}}\{\boldsymbol{\sigma}^{*1}, \boldsymbol{\sigma}^{*2}, \ldots, \boldsymbol{\sigma}^{*N}, \boldsymbol{\xi}^{*N+1}, \boldsymbol{\xi}^{*N+2}, \ldots, \boldsymbol{\xi}^{*2d+16}\} , \tag{3.6}$$

and

$$\boldsymbol{\sigma}_a \cdot \boldsymbol{\sigma}^{*b} = \delta_a^b , \qquad \boldsymbol{\sigma}_a \cdot \boldsymbol{\xi}^{*i} = 0 , \qquad \boldsymbol{\xi}_i \cdot \boldsymbol{\sigma}^{*a} = 0 , \qquad \boldsymbol{\xi}_i \cdot \boldsymbol{\xi}^{*j} = \delta_i^j . \tag{3.7}$$

In particular, we have the decomposition

$$\Gamma = \mathrm{Span}_{\mathbb{Z}}\{\boldsymbol{\sigma}^{*1}, \boldsymbol{\sigma}^{*2}, \ldots, \boldsymbol{\sigma}^{*N}\} + \Gamma_{\perp} , \quad \Gamma_{\perp} = \mathrm{Span}_{\mathbb{Z}}\{\boldsymbol{\xi}^{*N+1}, \boldsymbol{\xi}^{*N+2}, \ldots, \boldsymbol{\xi}^{*2d+16}\} . \tag{3.8}$$

### 3.1 Orbifold consistency conditions

Having described the most general symmetry we wish to consider, we now turn to gauging $G$, which is accomplished by introducing twisted sectors and taking a projection onto $G$–invariant states. In the heroic days the most concrete check of the consistency of such a procedure was obtained by calculating the partition function of the orbifolded theory and checking its modular invariance. In certain simple situations one could use modular orbits to construct a modular–invariant partition function and then interpret it as a sum over twisted sectors.[5] The more modern categorical perspective gives a more satisfying in-principle answer: we can think of the gauging as an introduction of a background field for the discrete group $G$, and there is a certain class $\omega \in H^3(G, \mathrm{U}(1))$, which must vanish if the sum over all gauge configurations is to be well-defined.[6]

We can follow a less technical approach because there is a particularly simple way to construct the unprojected twisted Hilbert spaces for each $\ell \in G$ based on spectral flow. At any point in the moduli space the CFT has a Kac-Moody symmetry $(\mathrm{U}(1)^{d+16})_{\mathrm{L}} \times (\mathrm{U}(1)^d)_{\mathrm{R}}$, which acts on the vertex operators $\mathcal{V}_{\boldsymbol{p}}$ labeled by $\boldsymbol{p} \in \Gamma$. For any $\ell \in G$ we can find a linear combination of the currents, with left- and right-moving components $J^\ell_{\mathrm{L}}$ and $J^\ell_{\mathrm{R}}$ such that the charge of $\mathcal{V}_{\boldsymbol{p}}$ with respect to this current is precisely $\sum_a \ell_a s_a \cdot \boldsymbol{p}$. We can use these currents to construct a twist field $\Sigma_\ell$, such that for every vertex operator $\mathcal{V}_{\boldsymbol{p}}$ in the original Hilbert space $\mathcal{H}_0$, we have

$$\mathcal{V}_{\boldsymbol{p}}(e^{2\pi i}z, e^{-2\pi i}\overline{z})\Sigma_\ell(0) = U(g, \boldsymbol{p})\mathcal{V}_{\boldsymbol{p}}(z, \overline{z})\Sigma_\ell(0) \ . \tag{3.9}$$

This is exactly the monodromy for the vertex operators in the twisted sector twisted by $\ell$, and we can therefore describe the twisted Hilbert space $\mathcal{H}_\ell$: it is isomorphic to $\mathcal{H}_0$, and the corresponding vertex operators are $\mathcal{V}_{\widetilde{\boldsymbol{p}}}$, where $\widetilde{\boldsymbol{p}} = \boldsymbol{p} + \sum_a \ell_a s_a$, and $\boldsymbol{p} \in \Gamma$. The spin and weight of each $\mathcal{V}_{\widetilde{\boldsymbol{p}}}$ are determined by (2.10) and (2.11) with the replacement $\boldsymbol{p} \to \widetilde{\boldsymbol{p}}$, so that we know the full twisted sector spectrum. We set

$$\Gamma^\ell = \{\boldsymbol{p} + \textstyle\sum_a \ell_a s_a \ \mid \ \boldsymbol{p} \in \Gamma\} \ . \tag{3.10}$$

With the unprojected twisted sectors in hand, it remains to choose a projection to invariant states, which we write in terms of a set of orbifold charges

$$Q^\ell_a : \Gamma^\ell \to \mathbb{Q} \ , \tag{3.11}$$

and we keep all operators with $\widetilde{\boldsymbol{p}} \in \Gamma^\ell$ satisfying $Q^\ell_a(\widetilde{\boldsymbol{p}}) \in \mathbb{Z}$ for all $a$. We constrain these charges by the following consistency conditions.

---

[5]A lucid early review of this can be found in [14]; a modern perspective is given in [1].

[6]A useful discussion and additional references can be found in [15], which also gives a generalization of the construction to higher dimensions.

1. The projection in the untwisted sector should be unmodified:

$$Q_a^0(\boldsymbol{p}) = s_a \cdot \boldsymbol{p} = \frac{\boldsymbol{\sigma}_a \cdot \boldsymbol{p}}{k_a} \ . \tag{3.12}$$

2. While $\Gamma^\ell$ is not an additive group, if $\widetilde{\boldsymbol{p}}_1 \in \Gamma^{\ell_1}$ and $\widetilde{\boldsymbol{p}}_2 \in \Gamma^{\ell_2}$, then $\widetilde{\boldsymbol{p}}_1 + \widetilde{\boldsymbol{p}}_2 \in \Gamma^{\ell_1+\ell_2}$. We demand that the OPE of the projected vertex operators is consistent with this additive structure, which implies

$$Q_a^\ell(\widetilde{\boldsymbol{p}}) = \frac{\boldsymbol{\sigma}_a \cdot \boldsymbol{p}}{k_a} - \sum_b \ell_b \zeta_{ba} \tag{3.13}$$

for some rational parameters $\zeta_{ba}$—these are called the "vacuum parameters" in [9, 10]. Notice that these are only meaningful modulo integers: a shift of $\zeta$ by an integral matrix leaves the orbifold projection unmodified.

3. We demand that each unprojected twisted sector Hilbert space is a representation of the gauge group $G$, which requires the $\zeta_{ba}$ to satisfy

$$\zeta_{ba} k_a \in \mathbb{Z} \tag{3.14}$$

for all $a$, $b$.

4. Finally, we require that every operator in the projected Hilbert space has integral spin. This is the only condition that is directly related to modular invariance of the orbifold partition function. A short calculation shows that this will hold if and only if the shift vectors $s_a$ and the vacuum parameters satisfy, for all $a$ and $b$ with $a < b$

$$2\zeta_{aa} + s_a \cdot s_a \in 2\mathbb{Z} \ , \qquad\qquad \zeta_{ab} + \zeta_{ba} + s_a \cdot s_b \in \mathbb{Z} \ . \tag{3.15}$$

These in turn imply the following conditions on the shift vectors for all $a$ and $b$ with $a < b$:

$$\boldsymbol{\sigma}_a . \boldsymbol{\sigma}_a \in 2k_a\mathbb{Z} \ , \qquad\qquad \boldsymbol{\sigma}_a . \boldsymbol{\sigma}_b \in k_a\mathbb{Z} \ . \tag{3.16}$$

It is not hard to show [10] that if (3.16) hold, then there is always a choice of the $\zeta$ such that (3.14) and (3.15) are obeyed. For example, we can set

$$\zeta_{aa} = -\frac{s_a . s_a}{2} \ ,$$
$$\zeta_{ab} = -\frac{k_a Y_{ab} s_a . s_b}{\gcd(k_a, k_b)} \ , \qquad a \neq b \ , \tag{3.17}$$

where the integers $Y_{ab}$, with $a \neq b$ satisfy

$$k_a Y_{ab} + k_b Y_{ba} = \gcd(k_a, k_b) \ . \tag{3.18}$$

One can take this a step further and show that the most general solution differs from this one by $\zeta \to \zeta + \widetilde{\zeta}$, where $\widetilde{\zeta}_{ab}$ obeys

$$\widetilde{\zeta}_{ab} + \widetilde{\zeta}_{ba} = 0 \ , \qquad\qquad \widetilde{\zeta}_{ba} k_a \in \mathbb{Z} \ . \tag{3.19}$$

This shift can be interpreted as a choice of discrete torsion.

To demonstrate that this procedure leads to a consistent orbifold, we show that the invariant lattice, which describes the projected operators, is even self-dual. We set

$$\Gamma_{\mathrm{inv}} = \cup_\ell \Gamma^\ell_{\mathrm{inv}} \ , \qquad\qquad \Gamma^\ell_{\mathrm{inv}} = \{\widetilde{\boldsymbol{p}} \in \Gamma^\ell \ | \ Q^\ell_a(\widetilde{\boldsymbol{p}}) \in \mathbb{Z} \quad \text{for all } a\} \ . \tag{3.20}$$

Using the distinguished basis described above, we see that

$$\Gamma_{\mathrm{inv}} = \mathrm{Span}_{\mathbb{Z}}\{t_1, \ldots, t_N\} + \Gamma^0_{\mathrm{inv}} \ , \tag{3.21}$$

where

$$\Gamma^0_{\mathrm{inv}} = \mathrm{Span}_{\mathbb{Z}}\{k_1 \boldsymbol{\sigma}^{*1}, k_2 \boldsymbol{\sigma}^{*2}, \ldots, k_N \boldsymbol{\sigma}^{*N}\} + \Gamma_\perp \ , \tag{3.22}$$

and

$$t_a = s_a + \sum_b \zeta_{ab} k_b \boldsymbol{\sigma}^{*b} \ . \tag{3.23}$$

Since $t_a \cdot t_a \in 2\mathbb{Z}$, $\Gamma_{\mathrm{inv}}$ is an even integral lattice, and therefore also $\Gamma_{\mathrm{inv}} \subseteq (\Gamma_{\mathrm{inv}})^*$. To show the opposite inclusion, suppose $v \in (\Gamma_{\mathrm{inv}})^*$. This holds if and only if

$$v = \sum_b S^b s_b + \sum_i X^i \boldsymbol{\xi}_i \ , \tag{3.24}$$

where the integers $S^b = \boldsymbol{\sigma}^{*b} \cdot v$ and $X^i = \boldsymbol{\xi}^{*i} \cdot v$ obey

$$t_a \cdot v = \sum_b S^b \left(s_a \cdot s_b + \zeta_{ab}\right) + \sum_i X^i s_a \cdot \boldsymbol{\xi}_i = T_a \in \mathbb{Z} \ . \tag{3.25}$$

Using this last relation, we can rewrite

$$\begin{aligned} v = {} & \sum_b S^b t_b + \sum_i X^i \left(\boldsymbol{\xi}_i - \sum_a (\boldsymbol{\xi}_i \cdot \boldsymbol{\sigma}_a) \boldsymbol{\sigma}^{*a}\right) \\ & + \sum_a \left(T_a - \sum_b S^b (s_a \cdot s_b + \zeta_{ab} + \zeta_{ba})\right) k_a \boldsymbol{\sigma}^{*a} \ . \end{aligned} \tag{3.26}$$

This shows $v \in \Gamma_{\mathrm{inv}}$ since each sum belongs separately to $\Gamma_{\mathrm{inv}}$ .

## 3.2 Relation to the "shifting method"

The "shifting method" is a procedure, introduced long ago—see especially [10] and [8], that produces an integral lattice $\Gamma_{\mathrm{inv}} = \Gamma(\{s_1, \ldots, s_N\}, \zeta)$ given an integral lattice $\Gamma$ and a choice of shift vectors and vacuum parameters just as in (3.21). When the shift vectors and vacuum parameters obey (3.14) and a weakened version of (3.15)

$$\zeta_{ab} + \zeta_{ba} + s_a \cdot s_b \in \mathbb{Z} \qquad \text{for all } a \text{ and } b , \tag{3.27}$$

$\Gamma_{\mathrm{inv}}$ is self-dual. This, and many additional results are obtained in [10]. Our discussion in the previous subsection gives a physical rationale for this construction when further restricted to even lattices: in this case the shifting method indeed produces the lattice associated to the general shift orbifold.

As we described above, once the shift vectors obey (3.16), there is always a choice of vacuum parameters—or, equivalently, an extension of the orbifold action to the twisted sectors—that leads to a consistent orbifold, and any two choices of $\zeta$ differ by a discrete torsion phase. A beautiful result of [10] shows that for any consistent shift orbifold $\Gamma_{\mathrm{inv}} = \Gamma(\{s_1, \ldots, s_N\}, \zeta)$ and a set of lattice vectors $\boldsymbol{x}_a \in \Gamma$, there is another consistent orbifold with $\Gamma'_{\mathrm{inv}} = \Gamma(\{s'_1, \ldots, s'_N\}, \zeta')$, where

$$s'_a = s_a + \boldsymbol{x}_a , \qquad\qquad \zeta'_{ba} = \zeta_{ba} - \boldsymbol{x}_b.(\boldsymbol{x}_a + s_a) , \tag{3.28}$$

and $\Gamma_{\mathrm{inv}} = \Gamma'_{\mathrm{inv}}$. Note this is not merely a lattice isomorphism but in fact a pointwise equality of the elements as embedded in $\mathbb{R}^{d,d+16}$. Thus, any two shift orbifolds related by (3.28) lead to the same orbifold CFT. In fact, it is always possible to choose the $\boldsymbol{x}_a$ lattice vectors so that $\zeta' = 0 \mod \mathbb{Z}$ [10]. To do so, we set

$$\boldsymbol{x}_a = \sum_c \zeta_{ac} k_c \boldsymbol{\sigma}^{*c} . \tag{3.29}$$

The condition (3.14) shows that these are lattice vectors, and

$$\zeta'_{ba} = -\boldsymbol{x}_b.\boldsymbol{x}_a \in \mathbb{Z} . \tag{3.30}$$

The new shift vectors now satisfy the simpler consistency condition

$$s'_a \cdot s'_b \in \mathbb{Z} . \tag{3.31}$$

## 3.3 The quantum symmetry

An orbifold theory is always equipped with a *quantum symmetry*, which is not present in the parent theory, and which, when gauged, leads back to the parent theory. This is familiar when the gauge group $G$ is abelian—see, for example, [14], but it can also be extended to more general $G$ by using categorical methods [15, 16]. The quantum symmetry for a shift orbifold with $s_a \cdot s_b \in \mathbb{Z}$ and $\zeta_{ab} = 0$ is another shift action, generated by the shift vectors

$\widetilde{s}^a = \widetilde{\boldsymbol{\sigma}}^a / k_a$, with $\widetilde{\boldsymbol{\sigma}}^a = k_a \boldsymbol{\sigma}^{*a} \in \Gamma_{\text{inv}}$. It is easy to see that this generates an action by the Pontryagin-dual group $\widetilde{G} \simeq G$ via

$$U(\widetilde{\ell}, \widetilde{\boldsymbol{p}}) = \exp\left\{ 2\pi \mathrm{i} \sum_a \widetilde{\ell}_a \widetilde{s}^a \cdot \widetilde{\boldsymbol{p}} \right\} , \qquad (3.32)$$

where $\widetilde{\ell} \in \widetilde{G}$. This action projects out the twisted sector states in $\Gamma_{\text{inv}}^\ell$ for $\ell \neq 0$ and introduces back the sectors generated by $\boldsymbol{\sigma}^{*a}$ projected out by the original orbifold.

## 3.4 Dualities among shift vectors

Consider a primitive lattice element $\boldsymbol{\sigma} \in \Gamma$ whose norm takes the form

$$\boldsymbol{\sigma} \cdot \boldsymbol{\sigma} = 2k\hat{k} , \qquad (3.33)$$

with two positive integers $k$ and $\hat{k}$. There are two possible cyclic orbifolds of the CFT using this $\boldsymbol{\sigma}$ allowed by the consistency conditions (3.16): a $\mathbb{Z}_k$ shift orbifold with $s = \boldsymbol{\sigma}/k$, and a $\mathbb{Z}_{\hat{k}}$ shift orbifold with $\hat{s} = \boldsymbol{\sigma}/\hat{k}$. Remarkably, the two orbifold CFTs are equivalent. The isomorphism $\varphi$ between the corresponding lattices is a reflection through the hyperplane orthogonal to $\boldsymbol{\sigma}$[7]

$$\varphi(\boldsymbol{p}) = \boldsymbol{p} - \frac{\boldsymbol{\sigma} \cdot \boldsymbol{p}}{k\hat{k}} \boldsymbol{\sigma} . \qquad (3.34)$$

Indeed, this reflection preserves the sublattice $\Gamma_\perp$ orthogonal to $\boldsymbol{\sigma}$, and acts on the remaining lattice vectors as

$$\varphi\left(k\boldsymbol{\sigma}^*\right) = -\frac{\boldsymbol{\sigma}}{\hat{k}} + \frac{\boldsymbol{\sigma} \cdot \boldsymbol{\sigma}}{2\hat{k}} \boldsymbol{\sigma}^* , \qquad \varphi\left(\frac{1}{k}\boldsymbol{\sigma} - \frac{\boldsymbol{\sigma} \cdot \boldsymbol{\sigma}}{2k} \boldsymbol{\sigma}^*\right) = -\hat{k}\boldsymbol{\sigma}^* . \qquad (3.35)$$

When the lattice $\Gamma$ is Euclidean, the above statement implies that every $\mathbb{Z}_k$ shift orbifold with shift vector $s$ admits a dual presentation as a $\mathbb{Z}_{\hat{k}}$ shift orbifold, where $\hat{k} = k\, s \cdot s / 2$. The associated shift vector is

$$\hat{s} = \frac{2\, s}{s \cdot s} . \qquad (3.36)$$

In particular, if the original shift vector satisfies $s \cdot s = 2/k$, then $\hat{s}$ belongs to $\Gamma$ and the orbifold CFT is isomorphic to the original one.

## 4 Heterotic shift orbifolds

Having described the most general shift orbifold of a Narain CFT, we now specialize to ten dimensions and return to the question raised in the introduction: which shift orbifolds interchange the two lattices $\Lambda_{16}$ and $\Lambda_8 + \Lambda_8$? A partial answer is provided at the end of the

---

[7]We thank G. Chenevier for this observation, which is found in proposition 3.1.14 of [17].

previous section: a $\mathbb{Z}_k$ orbifold with a shift vector $s$ obeying $s \cdot s = 2/k$ does not exchange the lattices. In the rest of this section we will provide an algorithm for the more general situation.

The existence of orbifolds that map one heterotic string to the other has been known since the seminal work of [18]. A classic example is the $\mathbb{Z}_2$ orbifold of the $\Lambda_8 + \Lambda_8$ theory with shift vector[8]

$$s = (0^7, 1, -1, 0^7) \ . \tag{4.1}$$

The projected untwisted Hilbert space consists of states with $\boldsymbol{p} \in (D_8 + D_8) \cup (D_8 + D_8 + \boldsymbol{w}_+)$, where $D_8 = \mathfrak{spin}(16)$ and $\boldsymbol{w}_+ = (\frac{1}{2}, \dots, \frac{1}{2})$. The two $D_8 + D_8$ factors are completed to $D_{16}$ by the addition of twisted states, and thus orbifolding leads us to the $\Lambda_{16}$ theory. The existence of such a shift orbifold should not be too surprising: it is a direct implication of a result from [9], which establishes that any two self-dual lattices of the same dimension and signature can be related by a rational shift.

More generally, starting with one of the two ten-dimensional heterotic theories and orbifolding, the quotient theory will, by construction, be associated to an Euclidean even self-dual lattice of rank 16, and this invariant lattice must then be isomorphic to either $\Lambda_8 + \Lambda_8$ or $\Lambda_{16}$. A direct approach to identify which possibility is realized would be to algorithmically search for a lattice isomorphism between the invariant lattice and $\Lambda_8 + \Lambda_8$ or $\Lambda_{16}$. There are well-known algorithms for solving this so-called lattice isomorphism problem [19], and given the rank of the lattices and their symmetries, these methods converge reasonably quickly. However, this answer is not entirely satisfactory, as it does not provide an explanation for why a given set of shift vectors $s_a = \boldsymbol{\sigma}_a / k_a$ would lead to an interchange of the two heterotic strings or not. Such an explanation is particularly desirable when the order of the orbifold group is large, and it becomes harder to directly identify the breaking patterns and the recombination with twisted states, as we did in the $\mathbb{Z}_2$ example above. For instance, the reader might be surprised that a $\mathbb{Z}_3$ shift orbifold of the $\mathrm{Spin}(32)/\mathbb{Z}_2$ theory with

$$\boldsymbol{\sigma} = (2, 1^8, 0^7) \quad \in \Lambda_{16} \tag{4.2}$$

gives the $(\mathrm{E}_8 \times \mathrm{E}_8) \rtimes \mathbb{Z}_2$ theory. Similarly, a $\mathbb{Z}_3$ shift orbifold of the $(\mathrm{E}_8 \times \mathrm{E}_8) \rtimes \mathbb{Z}_2$ string with

$$\boldsymbol{\sigma} = (\tfrac{1}{2}, -\tfrac{1}{2}^6, \tfrac{5}{2}, -2, 0^7) \quad \in \Lambda_8 + \Lambda_8 \tag{4.3}$$

sends it to the $\mathrm{Spin}(32)/\mathbb{Z}_2$ string. While the shift vectors (4.2) and (4.3) might seem exotic, the reader can convince themselves that in both cases, the Kac–Moody currents preserved in the untwisted sector generate a $\mathfrak{u}(9) + \mathfrak{spin}(14)$ algebra, which is just right to fit as a subalgebra of $\mathfrak{e}_8 + \mathfrak{e}_8$ and of $\mathfrak{spin}(32)$.

---

[8]To lighten notation, we denote lattice vectors by their components in the orthonormal basis $\{v_\alpha\}$ of $\mathbb{R}^{16}$ introduced in subsection 2.1.

As we will see, there exists a fairly direct method to determine whether a given $\mathbb{Z}_k$ shift orbifold interchanges the two heterotic strings or not.[9] To tackle this question, it will turn out fruitful, as an intermediate step, to consider the nine-dimensional heterotic string. Why this should be the case is already hinted by the expression for the $\mathbb{Z}_2$ shift vector (4.1): it coincides with the Wilson line considered in [20] in order to continuously interpolate between the two heterotic strings inside the moduli space $\mathfrak{M}_{1,17}$. We will make use of this insight to determine, in general, which shift orbifolds go from one lattice to the other. Along the way, we will prove an equivalence between decompactification limits of the nine-dimensional heterotic string and rationality of Wilson lines.

## 4.1 A nine-dimensional perspective

Consider a cyclic $\mathbb{Z}_k$ shift orbifold of the ten-dimensional heterotic string, as described in section 3. We denote by $\Lambda$ the lattice of the Narain theory—either $\Lambda_8 + \Lambda_8$ or $\Lambda_{16}$ depending on which string we are considering. The shift vector is $s = \boldsymbol{\sigma}/k$, where $\boldsymbol{\sigma} \in \Lambda$ is a lattice element. We can embed $\Lambda$ in a Lorentzian self-dual lattice $\Gamma = \Lambda_{1,1} + \Lambda$ associated to the nine-dimensional heterotic theory. As recalled in subsection 2.1, the corresponding CFT is parametrized by two moduli: a radius $r$ and a Wilson line parameter $A \in \Lambda_{\mathbb{R}} \simeq \mathbb{R}^{16}$, which enter in the expressions of left- and right-moving weights. Let us take the Wilson line to be equal to the shift vector:

$$A = \frac{\boldsymbol{\sigma}}{k} \ . \tag{4.4}$$

We wish to investigate the current algebra arising in (some) decompactification limits of this theory. The left-moving currents arise from vertex operators $\mathcal{V}_{\boldsymbol{p}}$ with $s(\boldsymbol{p}) = 1$ and $h_{\mathrm{R}}(\boldsymbol{p}) \to 0$ in the decompactification limit.

We first consider the large radius limit $r \to \infty$. The right-moving weight (2.11) of an operator with $\boldsymbol{p} = \mathrm{w}\boldsymbol{e} + \mathrm{n}\boldsymbol{e}^* + \boldsymbol{L}$ can only vanish in the limit if $\mathrm{w} = 0$, in which case the spin simplifies to $s(\boldsymbol{p}) = \frac{1}{2}\boldsymbol{L} \cdot \boldsymbol{L}$. In particular, for every root $\boldsymbol{L}$ of the lattice $\Lambda$ there is a corresponding spin-1 vertex operator with $\boldsymbol{p} = \boldsymbol{L}$ and

$$h_{\mathrm{R}}(\boldsymbol{p}) = \frac{(s \cdot \boldsymbol{L})^2}{4r^2} \xrightarrow[r\to\infty]{} 0 \ . \tag{4.5}$$

Thus, the algebra corresponding to $\Lambda$ emerges in the limit, and we recover the ten-dimensional parent theory. Light states with non-zero momentum number n give an additional tower $\mathrm{Span}_{\mathbb{Z}}\{\boldsymbol{e}^*\}$ of Kaluza–Klein modes, decoupled from the gauge sector.

We now examine the $r \to 0$ limit. As addressed in [4], taking $r \to 0$ does not always provide a decompactification limit of the heterotic moduli space: it can also lead to periodic or chaotic behavior, depending on the value of the parameter $A$. In our case, coming back to (2.11), we see that any $\boldsymbol{p}$ satisfying $\mathrm{n} = \frac{s \cdot s}{2}\mathrm{w} + s \cdot \boldsymbol{L}$ will give an asymptotically holomorphic

---

[9]The generalization to the abelian case $G = \mathbb{Z}_{k_1} \times \cdots \mathbb{Z}_{k_N}$ is straightforward: by the results reviewed in the previous section, such an orbifold can be performed iteratively, one cyclic factor $\mathbb{Z}_{k_a}$ at a time.

light state of spin $s(\boldsymbol{p}) = \frac{1}{2}(\boldsymbol{L}+\mathrm{w}s)\cdot(\boldsymbol{L}+\mathrm{w}s)$. Elements of the invariant lattice $\Lambda_{\mathrm{inv}} = \cup_{\ell=0}^{k-1}\Lambda_{\mathrm{inv}}^{\ell}$ provide us with such states. Indeed, to any $\widetilde{\boldsymbol{L}} = \boldsymbol{L} + \ell s \in \Lambda_{\mathrm{inv}}^{\ell}$ we can associate the lattice vector $\boldsymbol{p} = \ell\boldsymbol{e} + (s\cdot\boldsymbol{L} + \frac{\ell s\cdot s}{2})\boldsymbol{e}^* + \boldsymbol{L} \in \Gamma.$[10] In the $r \to 0$ limit, the corresponding vertex operator becomes holomorphic, as its right-moving weight is simply

$$h_{\mathrm{R}}(\boldsymbol{p}) = \frac{r^2\ell^2}{4} \xrightarrow[r\to 0]{} 0 \ . \tag{4.6}$$

Hence, to every root of the lattice $\Lambda_{\mathrm{inv}}$ we can assign an emergent holomorphic current in the limit $r \to 0$. In addition, there is a tower of light Kaluza–Klein modes coming from the null lattice points $\boldsymbol{p} \in \mathrm{Span}_{\mathbb{Z}}\{-k\boldsymbol{e} + \frac{\boldsymbol{\sigma}\cdot\boldsymbol{\sigma}}{2k}\boldsymbol{e}^* + \boldsymbol{\sigma}\},$[11] that are decoupled from the gauge sector. We can reasonably expect that this corresponds to a decompactification limit—in other words, there should exist some duality frame of the Narain CFT with moduli $(r', A') = \mu_g(r, A)$, in which the $r \to 0$ limit described above appears as a straightforward $r' \to \infty$ large volume limit. We will now show this is the case.

## 4.2 The heterotic moduli space

To study the $r \to 0$ limit of the 9-dimensional heterotic theory parametrized by the Wilson line (4.4), we can usefully apply the general lessons from [4]. For this purpose we first need a description of the moduli space $\mathfrak{M}_{1,17}$ of the Narain CFT. This space is given as the quotient

$$\mathfrak{M}_{1,17} = \mathrm{Gr}(1, 17)/\mathrm{O}(\Gamma) \ , \tag{4.7}$$

where the T-duality group $\mathrm{O}(\Gamma)$ is identified with the isometry group of the lattice $\Gamma$.[12] Three subgroups of $\mathrm{O}(\Gamma)$ will be relevant for our analysis.

1. *Weyl symmetries* are parametrized by an isometry $R \in \mathrm{O}(\Lambda)$ of the Euclidean lattice $\Lambda$ and act by

$$\varphi_g(\boldsymbol{p}) = \mathrm{w}\boldsymbol{e} + \mathrm{n}\boldsymbol{e}^* + R(\boldsymbol{L}) \ , \qquad\qquad \mu_g(r, A) = (r, R^{-1}(A)) \ . \tag{4.8}$$

2. *Wilson line shifts* are parametrized by a lattice vector $\boldsymbol{q} \in \Lambda$ and act by

$$\varphi_g(\boldsymbol{p}) = \mathrm{w}\boldsymbol{e} + (\mathrm{n} + \boldsymbol{q}\cdot\boldsymbol{L} - \tfrac{1}{2}\boldsymbol{q}\cdot\boldsymbol{q}\mathrm{w})\boldsymbol{e}^* + \boldsymbol{L} - \boldsymbol{q}\mathrm{w} \ , \qquad \mu_g(r, A) = (r, A - \boldsymbol{q}) \ . \tag{4.9}$$

---

[10]Using the description (3.21) of $\Lambda_{\mathrm{inv}}$, we can write $\widetilde{\boldsymbol{L}} = \boldsymbol{L}_0 + \ell t$, where $\boldsymbol{L}_0 \in \Lambda_{\mathrm{inv}}^0$ and $t = s + k\zeta\boldsymbol{\sigma}^*$. We then see that $s\cdot\boldsymbol{L} + \ell s\cdot s/2 = s\cdot\boldsymbol{L}_0 + \ell(\zeta + s\cdot s/2)$ is an integer, due to the orbifold consistency conditions (3.15).

[11]Note that $\boldsymbol{\sigma}\cdot\boldsymbol{\sigma} \in 2k\mathbb{Z}$ from the consistency conditions (3.16).

[12]In addition to the lattice action $\varphi_g : \Gamma \to \Gamma$, the duality $g \in \mathrm{O}(\Gamma)$ must also be specified by a choice of phase $U(g, \boldsymbol{p})$, as we recalled in subsection 2.3. For the generators of $\mathrm{O}(\Gamma)$ listed below, a consistent choice of such cocyles can be found in [13]. Since they play a spectator role in our study, we do not include them here.

3. The *factorized duality transformation* is the involution defined as[13]

$$\varphi_g(\boldsymbol{p}) = -\mathrm{n}\boldsymbol{e} - \mathrm{w}\boldsymbol{e}^* + \boldsymbol{L} \ , \qquad \mu_g(r, A) = \Big(\frac{r}{r^2 + \frac{1}{2}A \cdot A}, \frac{A}{r^2 + \frac{1}{2}A \cdot A}\Big) \ . \qquad (4.10)$$

We also introduce a basis for the lattices $\Lambda_{16}$ and $\Lambda_8 + \Lambda_8$, following the conventions of [12]. For $\Lambda_{16}$ we pick simple roots of the $\mathfrak{spin}(32)$ sublattice as

$$\begin{aligned}
\boldsymbol{\beta}_i &= (0^{i-1}, 1, -1, 0^{15-i}) \qquad \text{for } i = 1, \ldots, 15 \ , \\
\boldsymbol{\beta}_{16} &= (0^{14}, 1^2) \ .
\end{aligned} \qquad (4.11)$$

The lowest root is $\boldsymbol{\beta}_0 = (-1^2, 0^{14})$. For the lattice $\Lambda_8 + \Lambda_8$, our choice of simple roots $\boldsymbol{\alpha}_i$ is as follows:

$$\begin{aligned}
\boldsymbol{\alpha}_i &= (0^{i-1}, 1, -1, 0^{15-i}) \ , & \boldsymbol{\alpha}_i' &= (0^{15-i}, 1, -1, 0^{i-1}) \qquad \text{for } i = 1, \ldots, 6 \ , \\
\boldsymbol{\alpha}_7 &= (-1^2, 0^{14}) \ , & \boldsymbol{\alpha}_7' &= (0^{14}, 1^2) \ , \\
\boldsymbol{\alpha}_8 &= (\tfrac{1}{2}^8, 0^8) \ , & \boldsymbol{\alpha}_8' &= (0^8, -\tfrac{1}{2}^8) \ .
\end{aligned} \qquad (4.12)$$

The lowest roots of the two $\Lambda_8$ factors are $\boldsymbol{\alpha}_0 = (0^6, 1, -1, 0^8)$ and $\boldsymbol{\alpha}_0' = (0^8, 1, -1, 0^6)$.

We are now ready to describe the parameter space $\mathfrak{M}_{1,17}$, following [4]. Consider the Spin(32)/$\mathbb{Z}_2$ frame and decompose the Narain lattice as $\Gamma = \Lambda_{1,1} + \Lambda_{16}$. We construct the space $\mathfrak{M}_{1,17}$ by picking, inside $\mathrm{Gr}(1, 17)$, a fundamental region for the action of the T-duality group $\mathrm{O}(\Gamma)$. This is achieved in several steps. First, using a Wilson line shift by some lattice element $\boldsymbol{q} \in \mathfrak{spin}(32) \subset \Lambda_{16}$, we can bring the parameter $A$ to the maximal torus of Spin(32). Then we apply a Weyl symmetry that takes $A$ to a fundamental alcove of Spin(32)—this is a fundamental domain for the action of the Weyl group on the maximal torus. Making the standard choice for this alcove, we can impose the conditions $\boldsymbol{\beta}_i \cdot A \geq 0$ and $-\boldsymbol{\beta}_0 \cdot A \leq 1$ on the Weyl-transformed Wilson line parameter. The Wilson line shift by $\boldsymbol{w}_+$ can be combined with the Weyl transformation $R(a_1, \ldots, a_{16}) = (-a_{16}, \ldots, -a_1)$ to obtain a transformation that preserves the fundamental alcove of Spin(32). Making use of this duality, we can impose on $A$ the additional condition $\boldsymbol{w}_+ \cdot A \leq 2$. All together, the above requirements provide us with a description of the Spin(32)/$\mathbb{Z}_2$ moduli space as

$$\mathfrak{M}_{\mathrm{Spin}(32)/\mathbb{Z}_2} = \big\{ A \in \mathbb{R}^{16} \ \big| \ \boldsymbol{\beta}_{1,\ldots,16} \cdot A \geq 0 \ , \ -\boldsymbol{\beta}_0 \cdot A \leq 1 \ , \ \boldsymbol{w}_+ \cdot A \leq 2 \big\} \ . \qquad (4.13)$$

Taking into account the factorized duality, under which the quadratic combination $r^2 + \frac{1}{2}A \cdot A$ gets inverted, we arrive at the Narain moduli space in the Spin(32)/$\mathbb{Z}_2$ description:

$$\mathfrak{M}_{1,17} = \big\{ (r, A) \mid r \geq 0 \ , \ A \in \mathfrak{M}_{\mathrm{Spin}(32)/\mathbb{Z}_2} \ , \ r^2 + \tfrac{1}{2}A \cdot A \geq 1 \big\} \ . \qquad (4.14)$$

---

[13]We use a slightly different terminology than in [13], by combining the factorized duality of [13] with a circle reflection. This removes an unnecessary minus sign in the transformation law of $A$.

This representation of $\mathfrak{M}_{1,17}$ was first obtained, using generalized Dynkin diagram technology, in [21], where it was named the $\mathrm{Spin}(32)/\mathbb{Z}_2$ *chimney*.

The heterotic moduli space can also be described in a $(\mathrm{E}_8 \times \mathrm{E}_8) \rtimes \mathbb{Z}_2$ frame, where the Narain lattice is $\Gamma = \Gamma_{1,1} + \Lambda_8 + \Lambda_8$. In parallel with the previous discussion, we first bring $A$ in the maximal torus of $\mathrm{E}_8 \times \mathrm{E}_8$ using a Wilson line shift by some $\boldsymbol{q} \in \Lambda_8 + \Lambda_8$, and then to the fundamental alcove, in which $A$ has positive inner product with all the simple roots, and inner product with the highest root of each $\mathrm{E}_8$ factor bounded by 1. The $\mathbb{Z}_2$ outer automorphism of $\mathrm{E}_8 \times \mathrm{E}_8$ acts non-trivially on $A$, and we can use it to impose one last constraint, which we take to be $-\boldsymbol{\alpha}_0 \cdot A \geq -\boldsymbol{\alpha}'_0 \cdot A$. All together, we obtain the $(\mathrm{E}_8 \times \mathrm{E}_8) \rtimes \mathbb{Z}_2$ moduli space

$$\mathfrak{M}_{(\mathrm{E}_8 \times \mathrm{E}_8) \rtimes \mathbb{Z}_2} = \left\{ A \in \mathbb{R}^{16} \mid \boldsymbol{\alpha}_{1,\dots,16} \cdot A \geq 0 \ , \ -\boldsymbol{\alpha}'_0 \cdot A \leq -\boldsymbol{\alpha}_0 \cdot A \leq 1 \right\} \ . \tag{4.15}$$

The Narain moduli space is expressed in the $(\mathrm{E}_8 \times \mathrm{E}_8) \rtimes \mathbb{Z}_2$ frame by substituting $\mathfrak{M}_{(\mathrm{E}_8 \times \mathrm{E}_8) \rtimes \mathbb{Z}_2}$ for $\mathfrak{M}_{\mathrm{Spin}(32)/\mathbb{Z}_2}$ in (4.14).

Several comments are in order regarding the above description of $\mathfrak{M}_{1,17}$. First, inside the Narain moduli space the Wilson line parameter $A$ has its norm bounded by $A \cdot A \leq 2$. To see this, consider the $\mathrm{Spin}(32)/\mathbb{Z}_2$ frame. Since $\mathfrak{M}_{\mathrm{Spin}(32)/\mathbb{Z}_2}$ is a convex polytope, any point $A$ of maximal norm must be an extreme point, i.e. one that cannot be written as a convex combination of two distinct points. There are 17 possible candidates—the vertices of the polytope. A short computation shows that only one vertex reaches the maximal norm. We denote this distinguished point as $\mathfrak{m}_{\mathrm{S}}$. Its coordinates are given by

$$\mathfrak{m}_{\mathrm{S}} = (\tfrac{1}{2}^8, 0^8) \ . \tag{4.16}$$

The analysis is similar for the $(\mathrm{E}_8 \times \mathrm{E}_8) \rtimes \mathbb{Z}_2$ moduli space, and there we also find a unique point of maximal norm, namely

$$\mathfrak{m}_{\mathrm{E}} = (0^7, 1, -1, 0^7) \ . \tag{4.17}$$

Second, we can map between $(\mathrm{E}_8 \times \mathrm{E}_8) \rtimes \mathbb{Z}_2$ and $\mathrm{Spin}(32)/\mathbb{Z}_2$ frames via the Ginsparg map [20], which we denote by $g_{\mathrm{E} \to \mathrm{S}}$.[14] The equivalence of the two formulations can be traced to the uniqueness (up to isometries) of even-self dual lattices of a given indefinite signature. For the embeddings of $\Lambda_8 + \Lambda_8$ and $\Lambda_{16}$ in $\mathbb{R}^{16}$ chosen above, the duality map is specified by the lattice isomorphism $\varphi_{\mathrm{E} \to \mathrm{S}} : \Lambda_{1,1} + \Lambda_8 + \Lambda_8 \to \Lambda_{1,1} + \Lambda_{16}$, where

$$\begin{aligned} \varphi_{\mathrm{E} \to \mathrm{S}}(\boldsymbol{e}) &= 2(\boldsymbol{e} - \boldsymbol{e}^* - \mathfrak{m}_{\mathrm{S}}) + \mathfrak{m}_{\mathrm{E}} \ , \\ \varphi_{\mathrm{E} \to \mathrm{S}}(\boldsymbol{e}^*) &= -2(\boldsymbol{e} - \boldsymbol{e}^* - \mathfrak{m}_{\mathrm{S}}) \ , \\ \varphi_{\mathrm{E} \to \mathrm{S}}(\boldsymbol{L}) &= \boldsymbol{L} + 2(\mathfrak{m}_{\mathrm{E}} \cdot \boldsymbol{L})(\boldsymbol{e} - \boldsymbol{e}^* - \mathfrak{m}_{\mathrm{S}}) + (\mathfrak{m}_{\mathrm{S}} \cdot \boldsymbol{L})\boldsymbol{e}^* \ . \end{aligned} \tag{4.18}$$

The inverse isomorphism $\varphi_{\mathrm{S} \to \mathrm{E}}$ is obtained by exchanging the subscripts E and S in (4.18).

---

[14]This is the HE $\leftrightarrow$ HO map in [12], from which we borrow some of our conventions.

Let us emphasize that the Ginsparg map is not a lattice automorphism—in other words, the transformation $g_{\mathrm{E}\to\mathrm{S}}$ does not belong to the T-duality group $O(\Gamma)$. However, we can still use the Narain machinery introduced in section 2 to compute its action on the parameters. Denoting by $r_{\mathrm{E}}, A_{\mathrm{E}}$ and $r_{\mathrm{S}}, A_{\mathrm{S}}$ the moduli in, respectively, the $(\mathrm{E}_8 \times \mathrm{E}_8) \rtimes \mathbb{Z}_2$ and $\mathrm{Spin}(32)/\mathbb{Z}_2$ frames, we have $(r_{\mathrm{E}}, A_{\mathrm{E}}) = \mu_{g_{\mathrm{E}\to\mathrm{S}}}(r_{\mathrm{S}}, A_{\mathrm{S}})$, or more explicitly

$$r_{\mathrm{E}} = \frac{r_{\mathrm{S}}}{2\left(r_{\mathrm{S}}^2 + \frac{1}{2}(A_{\mathrm{S}} - \mathfrak{m}_{\mathrm{S}}) \cdot (A_{\mathrm{S}} - \mathfrak{m}_{\mathrm{S}})\right)} \ , \qquad A_{\mathrm{E}} = \frac{A_{\mathrm{S}} - \mathfrak{m}_{\mathrm{S}}}{2\left(r_{\mathrm{S}}^2 + \frac{1}{2}(A_{\mathrm{S}} - \mathfrak{m}_{\mathrm{S}}) \cdot (A_{\mathrm{S}} - \mathfrak{m}_{\mathrm{S}})\right)} + \mathfrak{m}_{\mathrm{E}} \ . \tag{4.19}$$

Here again, the inverse map is obtained by exchanging the E and S labels.

## 4.3 Decompactification limits and T-dualities

We now have the tools to resolve our initial puzzle of determining which ten-dimensional shift orbifolds switch between the two lattices $\Lambda_8 + \Lambda_8$ and $\Lambda_{16}$. As we have seen, answering this question is formally equivalent to identifying the $r \to 0$ limit of the nine-dimensional heterotic theory in the presence of a non-trivial Wilson line of the form (4.4). To tackle this latter problem, we can follow the hands-on method presented in [4].

Starting from a given point $(r, A)$ in $\mathfrak{M}_{1,17}$, we let the radius $r$ decrease until we reach the boundary $r^2 + \frac{1}{2}A \cdot A = 1$ of the Narain moduli space. To stay inside $\mathfrak{M}_{1,17}$, we must perform a T-duality—a factorized duality, followed by a lattice shift by some $\boldsymbol{q}$ and a Weyl symmetry $R$ that brings the new Wilson line parameter back inside the fundamental alcove of the gauge moduli space. In this T-dual frame, the new moduli

$$r' = \frac{r}{r^2 + \frac{1}{2}A \cdot A} \ , \qquad\qquad A' = \frac{R(A)}{r^2 + \frac{1}{2}A \cdot A} + R(\boldsymbol{q}) \tag{4.20}$$

depend on the initial radius $r$. In the $r \to 0$ limit, the dual radius goes to zero, while the dual Wilson line ends up at the value

$$R(\tfrac{2}{A \cdot A} A + \boldsymbol{q}) \ . \tag{4.21}$$

Our choice of $R$ and $\boldsymbol{q}$ is such that (4.21) is inside the fundamental alcove. At this point, there are three possibilities, depending on the value of $A$.

1. If $A$ is such that $\frac{2}{A \cdot A} A$ is a lattice vector, then $\boldsymbol{q} = -\frac{2}{A \cdot A} A$ and (4.21) vanishes. We perform an additional factorized duality. The new Wilson line parameter $A''$ goes to zero when $r \to 0$, while $r''$ runs off to infinity. Hence, we obtain a dual picture of the initial theory in which $r \to 0$ appears as a decompactification limit.

2. The second possibility is when $A \cdot A = 2$. As we saw in the previous subsection, this is only possible at one point of the moduli space, namely $\mathfrak{m}_{\mathrm{E}}$ for the $\mathfrak{M}_{(\mathrm{E}_8 \times \mathrm{E}_8) \rtimes \mathbb{Z}_2}$ moduli space and $\mathfrak{m}_{\mathrm{S}}$ for $\mathfrak{M}_{\mathrm{Spin}(32)/\mathbb{Z}_2}$. Let us generically denote this point, and the

corresponding fundamental alcove, by

$$\mathfrak{m} \ \in \mathfrak{M} \ . \tag{4.22}$$

When $A$ is equal to $\mathfrak{m}$, then $\boldsymbol{q} = 0$ and (4.21) equals $\mathfrak{m}$. We can use the Ginsparg map (4.18) to get to a dual frame, with moduli $r''$ and $A''$ obtained from (4.19). When $r \to 0$, the radius $r''$ goes to infinity: this is a decompactification limit, albeit of the other heterotic string.

3. Outside the two special cases above, (4.21) is a non-trivial vector in $\mathfrak{M}$. We are faced again with our initial question, since we now want to understand the limit $r' \to 0$ in the presence of the Wilson line (4.21). Hence, we can repeat the previous procedure: we apply a factorized duality, followed by a Wilson line shift and a Weyl reflection in order to keep our parameters inside the Narain moduli space.

**An iterative method**

Applying the above procedure to the Wilson line parameter (4.4) will identify the $r \to 0$ decompactification limit with one of the two ten-dimensional heterotic strings. In fact, we can forget about the nine-dimensional description and rephrase this method at the level of the lattice $\Lambda$. We start with a given shift vector $s = \boldsymbol{\sigma}/k$ associated with a cyclic shift orbifold of order $k$. Since translations of $s$ by lattice vectors do not affect the phase by which the orbifold acts on vertex operators, we can take $s$ to be in the maximal torus of the gauge group. Moreover, instead of $s$, we could pick any Weyl rotated shift vector $R(s)$—this amounts to working in a T-dual frame. Hence, without loss of generality, we can restrict to a shift vector inside the fundamental alcove:

$$s = \frac{\boldsymbol{\sigma}}{k} \ \in \mathfrak{M} \ . \tag{4.23}$$

We now define a sequence $\{s_i\}_{i \in \mathbb{N}}$ of vectors $s_i \in \mathfrak{M}$ as follows. We start with $s_0 = s$. At each step, if $s_i$ is non-zero, then we take $s_{i+1}$ to be a representative in the moduli space $\mathfrak{M}$ of the vector

$$\frac{2 s_i}{s_i \cdot s_i} \ . \tag{4.24}$$

If $s_i = 0$, we instead define $s_{i+1} = 0$. We claim that this sequence converges to a fixed point after a finite number of steps. In fact, for $i \geq k$ we either have $s_i = 0$ or $s_i = \mathfrak{m}$. In the first case, the orbifold theory is described by the same lattice $\Lambda$ as the parent theory. In the second case, the shift orbifold leads to an interchange of the two lattices $\Lambda_8 + \Lambda_8$ and $\Lambda_{16}$.

To prove this statement, we first observe that at each step we can write $s_i$ in the form

$$s_i = \frac{\boldsymbol{\sigma}_i}{k_i} \ , \tag{4.25}$$

where $k_\mathrm{i}$ is a positive integer and $\boldsymbol{\sigma}_\mathrm{i}$ is a lattice element satisfying the condition $\boldsymbol{\sigma}_\mathrm{i} \cdot \boldsymbol{\sigma}_\mathrm{i} \in 2k_\mathrm{i}\mathbb{Z}$. This is simply shown by induction. The initial case follows from the orbifold consistency conditions (3.16). Then at each step, we define[15]

$$k_{\mathrm{i}+1} = \frac{\boldsymbol{\sigma}_\mathrm{i} \cdot \boldsymbol{\sigma}_\mathrm{i}}{2k_\mathrm{i}} \ . \tag{4.26}$$

By the induction hypothesis, $k_{\mathrm{i}+1}$ is a positive integer. We can use it to write the norm of $s_\mathrm{i}$ as $s_\mathrm{i} \cdot s_\mathrm{i} = 2k_{\mathrm{i}+1}/k_\mathrm{i}$. By construction, $s_\mathrm{i}$ belongs to the moduli space $\mathfrak{M}$, and consequently its norm is bounded by $s_\mathrm{i} \cdot s_\mathrm{i} \leq 2$. Therefore, we must have

$$k_{\mathrm{i}+1} \leq k_\mathrm{i} \ , \tag{4.27}$$

and the equality $k_{\mathrm{i}+1} = k_\mathrm{i}$ holds if and only if $s_\mathrm{i} = \mathfrak{m}$. Since $s_{\mathrm{i}+1}$ represents the vector $\frac{2}{s_\mathrm{i} \cdot s_\mathrm{i}} s_\mathrm{i}$ in the fundamental domain, we can write it as $s_{\mathrm{i}+1} = R_\mathrm{i}(\frac{2}{s_\mathrm{i} \cdot s_\mathrm{i}} s_\mathrm{i} + \boldsymbol{q}_\mathrm{i})$, for some Weyl symmetry $R_\mathrm{i} \in \mathrm{O}(\Lambda)$ and a given lattice vector $\boldsymbol{q}_\mathrm{i} \in \Lambda$. This allows us to recast $s_{\mathrm{i}+1}$ as

$$s_{\mathrm{i}+1} = \frac{\boldsymbol{\sigma}_{\mathrm{i}+1}}{k_{\mathrm{i}+1}} \ , \qquad\qquad \boldsymbol{\sigma}_{\mathrm{i}+1} = R_\mathrm{i}(\boldsymbol{\sigma}_\mathrm{i} + k_{\mathrm{i}+1}\boldsymbol{q}_\mathrm{i}) \ . \tag{4.28}$$

Using (4.26), we readily compute $\boldsymbol{\sigma}_{\mathrm{i}+1} \cdot \boldsymbol{\sigma}_{\mathrm{i}+1} = 2k_{\mathrm{i}+1}(k_\mathrm{i} + \boldsymbol{\sigma}_\mathrm{i} \cdot \boldsymbol{q}_\mathrm{i} + \frac{1}{2}\boldsymbol{q}_\mathrm{i} \cdot \boldsymbol{q}_\mathrm{i}k_{\mathrm{i}+1})$. This shows that

$$\boldsymbol{\sigma}_{\mathrm{i}+1} \cdot \boldsymbol{\sigma}_{\mathrm{i}+1} \in 2k_{\mathrm{i}+1}\mathbb{Z} \ , \tag{4.29}$$

achieving the proof of the statement (4.25). The convergence of the sequence $\{s_\mathrm{i}\}$ follows: if at the i-th step the vector $s_\mathrm{i} = \boldsymbol{\sigma}_\mathrm{i}/k_\mathrm{i}$ is not equal to $0$ or $\mathfrak{m}$, then at the next step we have $s_{\mathrm{i}+1} = \boldsymbol{\sigma}_{\mathrm{i}+1}/k_{\mathrm{i}+1}$ with $k_{\mathrm{i}+1}$ a positive integer *strictly smaller* than $k_\mathrm{i}$. If the sequence did not converge, we would obtain an infinite sequence of positive integers with $k > k_1 > \cdots > k_\mathrm{i} > k_{\mathrm{i}+1} > \ldots$, which is of course impossible. Hence, we see that after a finite number of steps (smaller than $k$), $s_\mathrm{i}$ equals $0$ or $\mathfrak{m}$. The Weyl symmetries $R_\mathrm{i}$ and lattice elements $\boldsymbol{q}_\mathrm{i}$ allow us to reconstruct, in nine dimensions, a sequence of dualities leading to a frame in which $r \to 0$ appears as a decompactification limit.

### An aside: rational Wilson lines and decompactification

Using the above construction we can characterize for which values of $A$ the $r \to 0$ limit is a decompactification limit. This will happen if and only if the Wilson line parameter $A$ is *rational*—i.e. if it has rational coefficients when expanded in a basis of simple roots (or equivalently, of coweights).

When the sequence of T-dualities introduced at the beginning of this subsection converges, $A$ is related to $0$ or $\mathfrak{m}$ by a finite number of lattice vectors shifts and Weyl transformations. As a consequence, such an $A$ must be rational. Conversely, consider an arbitrary

---

[15]When $\boldsymbol{\sigma}_\mathrm{i} = 0$, we instead take $k_{\mathrm{i}+1} = 1$. Since $s_{\mathrm{i}+1} = 0$ the equality (4.25) is trivial in this case.

rational Wilson line

$$A = \sum_{i=1}^{16} \frac{m_i}{n_i} \boldsymbol{\alpha}_i \ , \tag{4.30}$$

where $m_i$ and $n_i$ are coprime integers for all $i = 1, \ldots, 16$. We define $k = \mathrm{lcm}(n_1, \ldots, n_{16})$ as the least common multiple of the integers in the denominators. In terms of $k$, the Wilson line parameter takes the form $A = \boldsymbol{\sigma}/k$ for some lattice vector $\boldsymbol{\sigma}$. Let us apply the algorithm introduced above to $A$. We obtain a sequence of Wilson line parameters $\{A_i\}_{i\in\mathbb{N}}$, related to the initial one by a chain of T-dualities parametrized by $\boldsymbol{q}_i$ and $R_i$.

In order to prove the convergence of such a sequence, we relied on the condition $\boldsymbol{\sigma} \cdot \boldsymbol{\sigma} \in 2k\mathbb{Z}$—it allowed us to define $k_1$ at the first step by the equation $\boldsymbol{\sigma} \cdot \boldsymbol{\sigma} = 2kk_1$. For a generic choice of rational Wilson line (4.30), this condition is not satisfied. Nevertheless, this is easily remedied by defining $k_1$ to be the numerator of $\frac{\boldsymbol{\sigma} \cdot \boldsymbol{\sigma}}{2k} \in \mathbb{Q}$ when expressed as an irreducible fraction. Doing so, $s_1$ can be recast in the form (4.25), and the corresponding lattice element $\boldsymbol{\sigma}_1$ satisfies the condition $\boldsymbol{\sigma}_1 \cdot \boldsymbol{\sigma}_1 \in 2k_1\mathbb{Z}$. The rest of the proof carries over, and the sequence of Wilson line converges to either 0 or $\mathfrak{m}$. This completes the analysis of $r \to 0$ limits of [4]: we see that in the presence of a rational Wilson line, the $r \to 0$ limit of the nine-dimensional heterotic theory is always a decompactification limit.

## 4.4 Cyclic shift orbifolds of low order

We can illustrate the preceding discussion by a simple example: a $\mathbb{Z}_5$ shift orbifold of the $\mathrm{Spin}(32)/\mathbb{Z}_2$ theory with

$$s_0 = (\tfrac{2}{5}^7, \tfrac{1}{5}^2, 0^7) \ . \tag{4.31}$$

It is not hard to check that this shift vector obeys the orbifold consistency conditions (3.16), and that it belongs to the fundamental domain $\mathfrak{M}_{\mathrm{Spin}(32)/\mathbb{Z}_2}$. At the first step, we compute $\frac{2}{s_0 \cdot s_0} s_0 = \frac{1}{3}(2^7, 1^2, 0^7) = \frac{1}{3}(-1^7, -2, 1, 0^7) - \boldsymbol{q}_1$, where $\boldsymbol{q}_1 = (-1^8, 0^8) \in \Lambda_{16}$. After a shift by $\boldsymbol{q}_1$ and a Weyl symmetry $R_1 \in \mathrm{O}(\Lambda_{16})$ (a combination of even sign changes and coordinate permutations), we arrive at

$$s_1 = (\tfrac{2}{3}, \tfrac{1}{3}^8, 0^7) \ . \tag{4.32}$$

Notice this is exactly the $\mathbb{Z}_3$ example we had considered in (4.2). We continue iterating this process and compute $\frac{2}{s_1 \cdot s_1} s_1 = \frac{1}{2}(2, 1^8, 0^7) = \frac{1}{2}(0, 1^7, -1, 0^7) - \boldsymbol{q}_2$ with $\boldsymbol{q}_2 = (-1, 0^7, -1, 0^7)$. We perform a lattice shift by $\boldsymbol{q}_2$, then act with a Weyl transformation $R_2$ (we can for example take $R_2$ to be the Weyl reflection associated to the root $\boldsymbol{q}_2$). We obtain

$$s_2 = (\tfrac{1}{2}^8, 0^8) \ . \tag{4.33}$$

Since we land on the fixed point $s_2 = \mathfrak{m}_s$, we conclude that the orbifold theory has to be the $(E_8 \times E_8) \rtimes \mathbb{Z}_2$ heterotic string.

For small values of $k$, we can make a more systematic study of cyclic $\mathbb{Z}_k$ shift orbifolds. The preceding discussion shows that cyclic shift orbifolds are classified by the set of points[16]

$$\mathfrak{M}_{\mathrm{Spin}(32)/\mathbb{Z}_2} \cap \frac{\Lambda_{16}}{k} \tag{4.34}$$

for the $\mathrm{Spin}(32)/\mathbb{Z}_2$ theory, and by

$$\mathfrak{M}_{(E_8 \times E_8) \rtimes \mathbb{Z}_2} \cap \frac{\Lambda_8 + \Lambda_8}{k} \tag{4.35}$$

for the $(E_8 \times E_8) \rtimes \mathbb{Z}_2$ theory. For low values of $k$ we can scan through (4.34) and (4.35) and apply the algorithm of subsection 4.3 to each shift vector. In this way we obtain, for a given order $k$, a classification of *inequivalent* $\mathbb{Z}_k$ shift orbifolds—we say that two orbifold theories are equivalent if they are related by an automorphism of the parent theory.

We give in tables 1 and 2 below the list of inequivalent cyclic shift orbifold of order $k \leq 5$. The shift vectors marked in color lead to an exchange of $\Lambda_8 + \Lambda_8$ and $\Lambda_{16}$. We also provide, for each orbifold, the sublattice $(\Lambda_{\mathrm{inv}}^0)_{\mathrm{roots}}$ spanned by the invariant untwisted roots. Note that, apart from the $\mathbb{Z}_2$ case, this sublattice is not of full rank. From the nine-dimensional perspective, the corresponding Wilson line leads to a gauge group with abelian factors.

**Table 1**: Cyclic $\mathbb{Z}_k$ shift orbifolds of the $\mathrm{Spin}(32)/\mathbb{Z}_2$ theory; highlighted orbifolds exchange the lattice.

$$k = 2$$

| $\sigma$ | $(\Lambda_{\mathrm{inv}}^0)_{\mathrm{roots}}$ |
|---|---|
| $(2, 0^{15})$ | $\mathfrak{spin}(32)$ |
| $(1^8, 0^8)$ | $2\,\mathfrak{spin}(16)$ |
| $(1^4, 0^{12})$ | $\mathfrak{spin}(8) + \mathfrak{spin}(24)$ |
| $\frac{1}{2}(1^{16})$ | $\mathfrak{su}(16)$ |

$$k = 3$$

| $\sigma$ | $(\Lambda_{\mathrm{inv}}^0)_{\mathrm{roots}}$ |
|---|---|
| $(2, 1^8, 0^7)$ | $\mathfrak{su}(9) + \mathfrak{spin}(14)$ |
| $(2, 1^2, 0^{13})$ | $\mathfrak{su}(3) + \mathfrak{spin}(26)$ |
| $\frac{1}{2}(3, 1^{14}, -1)$ | $\mathfrak{su}(15)$ |
| $(1^{12}, 0^4)$ | $\mathfrak{su}(12) + \mathfrak{spin}(8)$ |

---

[16]More precisely, they are classified by the subset of points satisfying the consistency condition $s \cdot s \in \frac{2}{k}\mathbb{Z}$.

| $\boldsymbol{\sigma}$ | $(\Lambda^0_{\mathrm{inv}})_{\mathrm{roots}}$ |
|---|---|
| $(1^6, 0^{10})$ | $\mathfrak{su}(6) + \mathfrak{spin}(20)$ |

$$k = 4$$

| $\boldsymbol{\sigma}$ | $(\Lambda^0_{\mathrm{inv}})_{\mathrm{roots}}$ |
|---|---|
| $\frac{1}{2}(7, 1^{14}, -1)$ | $\mathfrak{su}(16)$ |
| $(3, 1^{14}, -1)$ | $\mathfrak{su}(16)$ |
| $(3, 1^7, 0^8)$ | $\mathfrak{su}(8) + \mathfrak{spin}(16)$ |
| $\frac{1}{2}(5, 3^7, 1^7, -1)$ | $2\,\mathfrak{su}(8)$ |
| $\frac{1}{2}(5, 3^3, 1^{11}, -1)$ | $\mathfrak{su}(4) + \mathfrak{su}(12)$ |
| $(2^5, 1^4, 0^7)$ | $\mathfrak{su}(4) + \mathfrak{spin}(10) + \mathfrak{spin}(14)$ |
| $(2^4, 1^8, 0^4)$ | $\mathfrak{su}(8) + 2\,\mathfrak{spin}(8)$ |
| $(2^3, 1^4, 0^9)$ | $2\,\mathfrak{su}(4) + \mathfrak{spin}(18)$ |
| $(2^2, 1^8, 0^6)$ | $2\,\mathfrak{su}(2) + \mathfrak{su}(8) + \mathfrak{spin}(12)$ |
| $(2, 1^{12}, 0^3)$ | $\mathfrak{su}(4) + \mathfrak{su}(12)$ |
| $(2, 1^4, 0^{11})$ | $\mathfrak{su}(4) + \mathfrak{spin}(22)$ |
| $\frac{1}{2}(3^6, 1^{10})$ | $\mathfrak{su}(6) + \mathfrak{su}(10)$ |
| $\frac{1}{2}(3^2, 1^{14})$ | $\mathfrak{su}(2) + \mathfrak{su}(14)$ |
| $(1^{15}, -1)$ | $\mathfrak{su}(16)$ |
| $(1^8, 0^8)$ | $\mathfrak{su}(8) + \mathfrak{spin}(16)$ |

$$k = 5$$

| $\boldsymbol{\sigma}$ | $(\Lambda^0_{\mathrm{inv}})_{\mathrm{roots}}$ |
|---|---|
| $(4, 1^{14}, 0)$ | $\mathfrak{su}(16)$ |
| $\frac{1}{2}(7, 3^7, 1^8)$ | $2\,\mathfrak{su}(8)$ |
| $(3, 2^4, 1^5, 0^6)$ | $2\,\mathfrak{su}(5) + \mathfrak{spin}(12)$ |
| $(3, 2^3, 1^9, 0^3)$ | $2\,\mathfrak{su}(4) + \mathfrak{su}(9)$ |
| $(3, 2^2, 1^{13})$ | $\mathfrak{su}(3) + \mathfrak{su}(13)$ |
| $(3, 2^2, 1^3, 0^{10})$ | $2\,\mathfrak{su}(3) + \mathfrak{spin}(20)$ |
| $(3, 1^{11}, 0^4)$ | $\mathfrak{su}(11) + \mathfrak{spin}(8)$ |
| $(3, 1, 0^{14})$ | $\mathfrak{spin}(28)$ |
| $\frac{1}{2}(5^4, 3, 1^{10}, -1)$ | $\mathfrak{su}(11) + \mathfrak{spin}(8)$ |
| $\frac{1}{2}(5^2, 3^7, 1^6, -1)$ | $2\mathfrak{su}(2) + 2\,\mathfrak{su}(7)$ |
| $\frac{1}{2}(5, 3^5, 1^9, -1)$ | $\mathfrak{su}(6) + \mathfrak{su}(10)$ |
| $(2^7, 1^2, 0^7)$ | $\mathfrak{su}(2) + \mathfrak{su}(7) + \mathfrak{spin}(14)$ |
| $(2^6, 1^6, 0^4)$ | $2\,\mathfrak{su}(6) + \mathfrak{spin}(8)$ |
| $(2^5, 1^{10}, 0)$ | $\mathfrak{su}(5) + \mathfrak{su}(10)$ |
| $(2^5, 0^{11})$ | $\mathfrak{su}(5) + \mathfrak{spin}(22)$ |
| $(2^4, 1^4, 0^8)$ | $2\,\mathfrak{su}(4) + \mathfrak{spin}(16)$ |

| $\sigma$ | $(\Lambda_{\mathrm{inv}}^0)_{\mathrm{roots}}$ |
|---|---|
| $(2^3, 1^8, 0^5)$ | $\mathfrak{su}(3) + \mathfrak{su}(8) + \mathfrak{spin}(10)$ |
| $(2^2, 1^{12}, 0^2)$ | $3\,\mathfrak{su}(2) + \mathfrak{su}(12)$ |
| $(2^2, 1^2, 0^{12})$ | $2\,\mathfrak{su}(2) + \mathfrak{spin}(24)$ |
| $(2, 1^6, 0^9)$ | $\mathfrak{su}(6) + \mathfrak{spin}(18)$ |
| $\frac{1}{2}(3^{13}, 1^2, -1)$ | $\mathfrak{su}(3) + \mathfrak{su}(13)$ |
| $\frac{1}{2}(3^8, 1^8)$ | $2\,\mathfrak{su}(8)$ |
| $\frac{1}{2}(3^3, 1^{12}, -1)$ | $\mathfrak{su}(3) + \mathfrak{su}(13)$ |
| $(1^{10}, 0^6)$ | $\mathfrak{su}(10) + \mathfrak{spin}(12)$ |

**Table 2**: Cyclic $\mathbb{Z}_k$ shift orbifolds of the $(\mathrm{E}_8 \times \mathrm{E}_8) \rtimes \mathbb{Z}_2$ theory; highlighted orbifolds exchange the lattice.

$$k = 2$$

| $\sigma$ | $(\Lambda_{\mathrm{inv}}^0)_{\mathrm{roots}}$ |
|---|---|
| $(0^7, 2, 0^8)$ | $\mathfrak{spin}(16) + \mathfrak{e}_8$ |
| $(0^7, 2, -2, 0^7)$ | $2\,\mathfrak{spin}(16)$ |
| $(0^6, -1, 1, -1, 1, 0^6)$ | $2\,\mathfrak{su}(2) + 2\,\mathfrak{e}_7$ |

$$k = 3$$

| $\sigma$ | $(\Lambda_{\mathrm{inv}}^0)_{\mathrm{roots}}$ |
|---|---|
| $\frac{1}{2}(1, -1^6, 5, -4, 0^7)$ | $\mathfrak{su}(9) + \mathfrak{spin}(14)$ |
| $(0^7, 2, -1, 1, 0^6)$ | $\mathfrak{spin}(14) + \mathfrak{e}_7$ |
| $(0^5, -1^2, 2, 0^8)$ | $\mathfrak{su}(3) + \mathfrak{e}_6 + \mathfrak{e}_8$ |
| $(0^5, -1^2, 2, -2, 1^2, 0^5)$ | $2\,\mathfrak{su}(3) + 2\,\mathfrak{e}_6$ |

$$k = 4$$

| $\sigma$ | $(\Lambda_{\mathrm{inv}}^0)_{\mathrm{roots}}$ |
|---|---|
| $\frac{1}{2}(1, -1^6, 5, 0^8)$ | $\mathfrak{su}(8) + \mathfrak{e}_8$ |
| $\frac{1}{2}(1, -1^6, 5, -5, 1^6, -1)$ | $2\,\mathfrak{su}(8)$ |
| $\frac{1}{2}(0^7, 8, -5, 1^6, -1)$ | $\mathfrak{su}(8) + \mathfrak{spin}(16)$ |
| $(0^7, 2, -2, 0^7)$ | $2\,\mathfrak{spin}(14)$ |
| $(0^6, -1, 3, -2, 1^2, 0^5)$ | $2\,\mathfrak{su}(2) + \mathfrak{spin}(12) + \mathfrak{e}_6$ |
| $\frac{1}{2}(0^6, -2, 6, -7, 1^7)$ | $2\,\mathfrak{su}(2) + \mathfrak{su}(8) + \mathfrak{spin}(12)$ |
| $\frac{1}{2}(0^6, -4, 4, -5, 1^6, -1)$ | $\mathfrak{su}(2) + \mathfrak{su}(8) + \mathfrak{e}_7$ |
| $(0^5, -1^2, 2, -1, 1, 0^6)$ | $\mathfrak{su}(2) + \mathfrak{e}_6 + \mathfrak{e}_7$ |
| $(0^4, -1^3, 3, -2, 0^7)$ | $\mathfrak{su}(4) + \mathfrak{spin}(10) + \mathfrak{spin}(14)$ |

| $\boldsymbol{\sigma}$ | $(\Lambda^0_{\mathrm{inv}})_{\mathrm{roots}}$ |
|---|---|
| $(0^4, -1^3, 3, -3, 1^3, 0^4)$ | $2\,\mathfrak{su}(4) + 2\,\mathfrak{spin}(10)$ |
| $\frac{1}{2}(-1^7, 7, -2, 2, 0^6)$ | $\mathfrak{su}(2) + \mathfrak{su}(8) + \mathfrak{e}_7$ |

$$k = 5$$

| $\boldsymbol{\sigma}$ | $(\Lambda^0_{\mathrm{inv}})_{\mathrm{roots}}$ |
|---|---|
| $\frac{1}{2}(1, -1^6, 9, -5, 1^6, -1)$ | $2\,\mathfrak{su}(8)$ |
| $\frac{1}{2}(1, -1^6, 5, -2, 2, 0^6)$ | $\mathfrak{su}(8) + \mathfrak{e}_7$ |
| $\frac{1}{2}(1, -1^5, -3, 7, -7, 1^7)$ | $2\,\mathfrak{su}(2) + 2\,\mathfrak{su}(7)$ |
| $(0^7, 4, -2, 0^7)$ | $2\,\mathfrak{spin}(14)$ |
| $\frac{1}{2}(0^7, 8, -7, 1^7)$ | $\mathfrak{su}(2) + \mathfrak{su}(7) + \mathfrak{spin}(14)$ |
| $(0^6, -1, 3, 0^8)$ | $\mathfrak{spin}(12) + \mathfrak{e}_8$ |
| $(0^6, -1, 3, -3, 1, 0^6)$ | $2\,\mathfrak{spin}(12)$ |
| $(0^6, -2, 2, -1, 1, 0^6)$ | $2\,\mathfrak{e}_7$ |
| $(0^5, -1^2, 4, -3, 1^3, 0^4)$ | $2\,\mathfrak{su}(3) + 2\,\mathfrak{spin}(10)$ |
| $(0^5, -1^2, 2, -2, 0^7)$ | $\mathfrak{su}(2) + \mathfrak{spin}(14) + \mathfrak{e}_6$ |
| $(0^5, -1, -2, 3, -2, 1^2, 0^5)$ | $2\,\mathfrak{su}(2) + 2\,\mathfrak{e}_6$ |
| $(0^4, -1^3, 3, -2, 2, 0^6)$ | $\mathfrak{su}(3) + \mathfrak{spin}(10) + \mathfrak{e}_7$ |
| $\frac{1}{2}(0^4, -2^3, 6, -5, 1^6, -1)$ | $\mathfrak{su}(3) + \mathfrak{su}(8) + \mathfrak{spin}(10)$ |
| $(0^3, -1^4, 4, 0^8)$ | $2\,\mathfrak{su}(5) + \mathfrak{e}_8$ |
| $(0^3, -1^4, 4, -3, 1, 0^6)$ | $2\,\mathfrak{su}(5) + \mathfrak{spin}(12)$ |
| $(0^3, -1^4, 4, -4, 1^4, 0^3)$ | $4\,\mathfrak{su}(5)$ |
| $\frac{1}{2}(-1^7, 7, -4, 2^2, 0^5)$ | $2\,\mathfrak{su}(2) + \mathfrak{su}(7) + \mathfrak{e}_6$ |

## 5 Shift orbifolds of Niemeier CFTs

In this final section we generalize some of our previous constructions to higher rank lattices and show that they lead to a systematic understanding of shift orbifolds of the Leech CFT.

One of the most interesting results of [10] is that any two even self-dual lattices of the same rank and signature can always be connected by the shifting method. Euclidean even self-dual lattices exist in any dimension divisible by 8. The lattice $\Lambda_8$ is the unique example in dimension 8. The statement of [10] becomes non-trivial in dimension 16, where there are two possibilities: $\Lambda_8 + \Lambda_8$ and $\Lambda_{16}$. We confirmed in section 4 the existence of shift orbifolds between the corresponding Narain theories. The next simplest examples can be found in dimension 24.

Euclidean even self-dual lattices of rank 24 have been classified in [22] and are known as Niemeier lattices. The list comprises twenty-four distinct lattices, including the famous example discovered by Leech in [23]. To every Niemeier lattice $\Lambda$ we can associate a conformal

field theory: the free theory of twenty-four holomorphic chiral bosons compactified on $\Lambda$. These theories appear in the Schellekens classification of meromorphic $c = 24$ CFTs [24].

Shift orbifolds of Niemeier CFTs, or closely related constructions disguised under various names, have been used to great effect in the physics literature. Several constructions of the Leech lattice—including the "holy constructions" of [25]—find a simple interpretation in the orbifold context, as we will describe below. CFT generalizations of such constructions have been used in the study of vertex operator algebras, and involve the so-called "generalized deep holes" of the Leech vertex operator algebra [26–28]. A recent application to constructions of isolated CFTs and heterotic islands is given in [29], and this reference also includes an overview of some additional appearances of the Leech lattice in physics.

Our aim will be to generalize methods from the previous section to the $c = 24$ Niemeier theories. In particular, we will obtain a pragmatic answer to the following question: when performing a cyclic shift orbifold of the Leech CFT, which Niemeier theory does one obtain? To frame the question, we first review some well-known results from the theory of even self-dual lattices, emphasizing the connections to shift orbifolds along the way.

## 5.1 Niemeier lattices

There are twenty-four even self-dual lattices with positive definite signature and rank 24. Any such lattice $\Lambda$ is fully characterized by its root system

$$\{\, \boldsymbol{L} \in \Lambda \mid \boldsymbol{L} \cdot \boldsymbol{L} = 2 \,\} \ . \tag{5.1}$$

The Leech lattice $\Lambda_{\mathrm{L}}$ is the unique Niemeier lattice without roots. In the other twenty-three cases the roots of $\Lambda$ span a sublattice

$$\mathfrak{g} = \mathfrak{g}_1 + \mathfrak{g}_2 + \cdots + \mathfrak{g}_M \ , \tag{5.2}$$

where each factor denotes the root lattice associated to a simply laced Dynkin diagram. This sublattice has full rank $\mathrm{rk}\,\mathfrak{g}_1 + \cdots + \mathrm{rk}\,\mathfrak{g}_M = 24$, and all of its components share the same Coxeter number $h$. The lattice $\Lambda$ can be reconstructed by gluing to $\mathfrak{g}$ appropriate elements of the discriminant group $\mathfrak{g}^* / \mathfrak{g}$.[17] We denote the corresponding Niemeier lattice by $\Lambda_{\mathfrak{g}}$. The gluing construction does not add any new root to the lattice, so the total number of roots of $\Lambda_{\mathfrak{g}}$ is equal to $24\,h$. The twenty-four Niemeier lattices are the nodes in figure 1 below.

For each Niemeier lattice with roots $\Lambda_{\mathfrak{g}}$, the Weyl vector $\boldsymbol{\rho}$, defined as half the sum of all positive roots, is an element of the lattice $\Lambda_{\mathfrak{g}}$ [31] with norm

$$\boldsymbol{\rho} \cdot \boldsymbol{\rho} = 2h(h + 1) \ . \tag{5.3}$$

The inner product of $\boldsymbol{\rho}$ with any root of $\Lambda_{\mathfrak{g}}$ is equal to the height of the root, and any positive

---

[17]We will not need the detailed form of these glue vectors. Their expression can be found in the classic reference [30].

root $\boldsymbol{\alpha}$ satisfies

$$\boldsymbol{\rho} \cdot \boldsymbol{\alpha} \in \{1, 2, \ldots, h-1\} \ . \tag{5.4}$$

**The Borcherds construction**

Starting from any of the twenty-three Niemeier lattices with roots $\Lambda_{\mathfrak{g}}$, the Weyl vector can be used to construct the Leech lattice as a cyclic shift orbifold of $\Lambda_{\mathfrak{g}}$ of order $h$—the Coxeter number of $\Lambda_{\mathfrak{g}}$. The twenty-three resulting descriptions of the Leech lattice are known as its "holy constructions" [25, 31], and each can be understood as a $\mathbb{Z}_h$ shift orbifold with shift vector $s = \frac{\rho}{h}$, which, as a result of (5.3), satisfies the orbifold consistency condition $s \cdot s \in \frac{2}{h}\mathbb{Z}$. Setting $\zeta = -1/h$, the invariant lattice takes the form

$$(\Lambda_{\mathfrak{g}})_{\text{inv}} = \left\{ \boldsymbol{L} + \tfrac{\ell}{h}\boldsymbol{\rho} \ \middle| \ \boldsymbol{L} \in \Lambda_{\mathfrak{g}} \ , \ \ell = 0, 1, \ldots, h-1 \ , \ \boldsymbol{L} \cdot \boldsymbol{\rho} \in h\mathbb{Z} - \ell \right\} \ . \tag{5.5}$$

The roots of the Niemeier lattice do not survive the projection to $\mathbb{Z}_h$-invariant states, due to the property (5.4), and the twisted sectors do not contain any root; since $(\Lambda_{\mathfrak{g}})_{\text{inv}}$ is an even-self dual lattice of rank 24 without roots, it must be isomorphic to the Leech lattice.

**Further relations between Niemeier lattices**

As we already mentioned, any two Niemeier lattices can be connected by a shift orbifold [10]. This statement is closely related to the theory of Kneser neighbors [32]. We say that two even self-dual lattices $\Lambda_1$ and $\Lambda_2$ are $k$-neighbors (with $k$ a prime number) if their intersection

$$\Lambda_1 \cap \Lambda_2 \tag{5.6}$$

is a full rank sublattice of index $k$ in $\Lambda_1$ (and therefore in $\Lambda_2$). When $\Lambda_1$ and $\Lambda_2$ are related by a $\mathbb{Z}_k$ shift orbifold, with $\Lambda_2 = (\Lambda_1)_{\text{inv}}$, it is easy to check that their intersection $\Lambda_1 \cap \Lambda_2 = (\Lambda_1)^0_{\text{inv}}$ has index $k$ in both of them. Conversely, when $\Lambda_1$ and $\Lambda_2$ are $k$-neighbors, we can pick any non-trivial element $s$ inside $\Lambda_2/(\Lambda_1 \cap \Lambda_2)$, and a cyclic orbifold of the $\Lambda_1$ theory with shift vector $s$ leads to $\Lambda_2$.

The study of Kneser neighboring has been foundational for investigations of even self-dual lattices. Kneser showed in [32] that any two even self-dual lattices of the same rank can be related by a $\mathbb{Z}_2 \times \cdots \times \mathbb{Z}_2$ shift orbifold (provided we allow enough cyclic factors). The 2-neighborhood graph of Niemeier lattices, plotted figure in 1, assigns a node to each Niemeier lattice, with a line drawn between two nodes if the corresponding lattices can be related by a $\mathbb{Z}_2$ shift orbifold. This graph was originally computed in [30]. For higher values of $k$, neighbors of Niemeier lattices have been extensively studied by Chenevier and Lannes [17, 33], among the results obtained there, the authors obtain a necessary condition for a Niemeier lattice $\Lambda$ to have the Leech lattice as a $k$-neighbor: $\Lambda$ can be connected to the Leech lattice by a $\mathbb{Z}_k$

shift orbifold only if

$$k \geq h \, , \tag{5.7}$$

where $h$ is the Coxeter number of $\Lambda$. Note that the shift orbifolds associated to the Borcherds construction saturate the inequality. In the following section, we will obtain an alternative proof of this bound by studying shift orbifolds of the Leech theory.

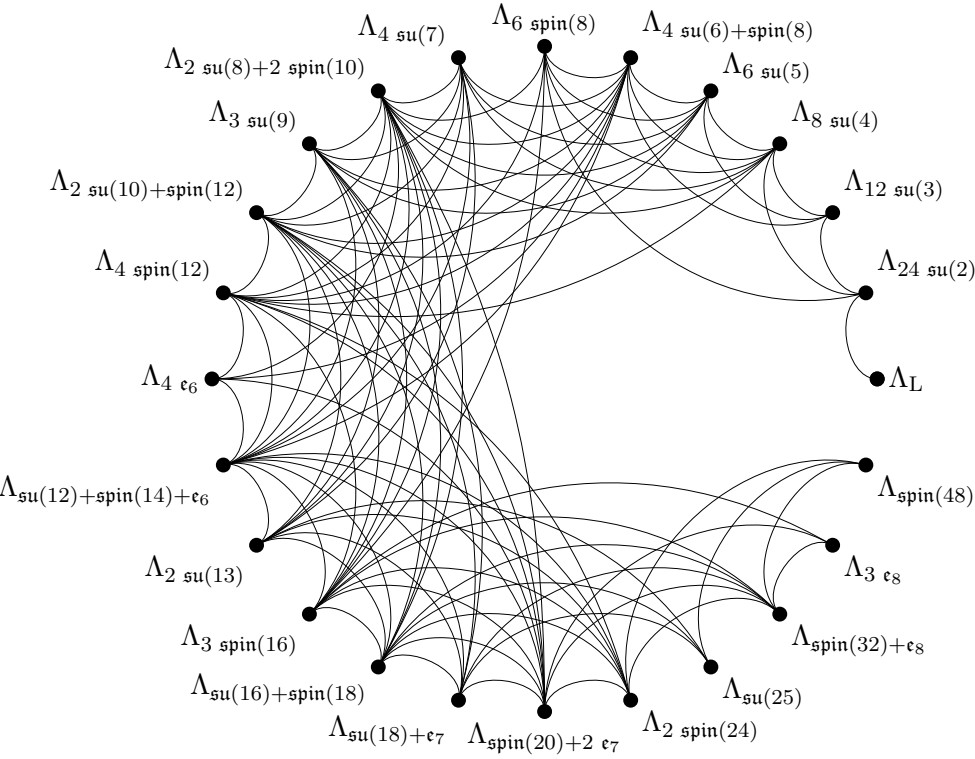

**Figure 1**. The 2-neighborhood graph of Niemeier lattices.

## 5.2    Orbifolds of the Leech CFT

We now turn our attention to the Leech CFT and its shift orbifolds. We saw in the previous section that the Leech theory can be obtained as a cyclic orbifold of any of the other Niemeier CFTs, using the Weyl vector of the Niemeier lattice. Following the discussion in section 3.3, the corresponding quantum symmetries provide us with twenty-three shift orbifolds of the Leech CFT, leading to distinct $c = 24$ Niemeier theories.

Our aim will be to characterize, given a $\mathbb{Z}_k$ shift symmetry of the Leech CFT, which of the twenty-four Niemeier CFTs describes the orbifold theory. For this purpose it is useful to introduce additional lattice machinery. Given a Euclidean lattice $\Lambda$, its Voronoi cell $V_\Lambda$ is the

convex polytope in $\Lambda_{\mathbb{R}}$ defined as

$$V_\Lambda = \left\{ v \in \Lambda_{\mathbb{R}} \mid v \cdot v \leq (v - \boldsymbol{L}) \cdot (v - \boldsymbol{L}) \quad \text{for all } \boldsymbol{L} \in \Lambda \right\} . \tag{5.8}$$

The Voronoi cell contains all the points which are closer to the origin than to any other lattice point, and it provides a fundamental region for the lattice $\Lambda$ that is preserved by lattice automorphisms: $R(V_\Lambda) = V_\Lambda$ for any $R \in O(\Lambda)$. A point whose distance from the lattice is a local maximum is a vertex of $V_\Lambda$ (or its translate by a lattice element), and is called a *hole*. The points which are the furthest away from the lattice are called *deep holes*, and their distance from the lattice is the covering radius $r_\Lambda$:[18]

$$r_\Lambda = \sup \left\{ \sqrt{v \cdot v} \mid v \in V_\Lambda \right\} . \tag{5.9}$$

The lattice points located at distance $r_\Lambda$ of a deep hole are the vertices of this deep hole.

We already encountered two examples of Voronoi cells in our description of the heterotic moduli space in section 4.2 :

$$V_{\Lambda_8 + \Lambda_8} = \bigcup_{R \in O(\Lambda_8 + \Lambda_8)} R\left(\mathfrak{M}_{(E_8 \times E_8) \rtimes \mathbb{Z}_2}\right) , \qquad V_{\Lambda_{16}} = \bigcup_{R \in O(\Lambda_{16})} R\left(\mathfrak{M}_{\mathrm{Spin}(32)/\mathbb{Z}_2}\right) . \tag{5.10}$$

The deep holes of $\Lambda_8 + \Lambda_8$ which share a vertex at the origin are the Weyl images of $\mathfrak{m}_E$, while the deep holes of $\Lambda_{16}$ near the origin are the Weyl images of $\mathfrak{m}_S$. The covering radius of both lattices is equal to $r_{\Lambda_8 + \Lambda_8} = r_{\Lambda_{16}} = \sqrt{2}$. The above description of the Voronoi cell of a lattice as a the union of all images of a simplex $\mathfrak{M}$ under the Weyl group is common to all root lattices, and the region $\mathfrak{M}$ is called a fundamental simplex. We refer the reader to [34, 35] for additional details.

The deep holes of the Leech lattice $\Lambda_L$ exhibit a rich structure uncovered in [36]. The covering radius of $\Lambda_L$ is $r_{\Lambda_L} = \sqrt{2}$, and hence any deep hole $\mathfrak{m}$ near the origin satisfies

$$\mathfrak{m} \cdot \mathfrak{m} = 2 . \tag{5.11}$$

For a given deep hole $\mathfrak{m}$, its vertices (i.e. the lattice points $\boldsymbol{\nu} \in \Lambda_L$ satisfying $(\boldsymbol{\nu} - \mathfrak{m}) \cdot (\boldsymbol{\nu} - \mathfrak{m}) = 2$) satisfy remarkable properties. The distance between two distinct vertices $\boldsymbol{\nu}$ and $\boldsymbol{\nu}'$ can only take the following values:

$$(\boldsymbol{\nu}' - \boldsymbol{\nu}) \cdot (\boldsymbol{\nu}' - \boldsymbol{\nu}) \in \{4, 6, 8\} . \tag{5.12}$$

To any deep hole $\mathfrak{m}$, we can assign a *hole diagram* as follows. We label the vertices of $\mathfrak{m}$ as $\boldsymbol{\nu}_i$, where the index $i$ runs over the total number of vertices. We associate a node of the diagram

---

[18]This is the smallest radius such that spheres centered at lattice points cover the whole vector space.

to each vertex. The number of lines between two nodes is given by

$$N_{ij} = \tfrac{1}{2}(\boldsymbol{\nu}_i - \boldsymbol{\nu}_j) \cdot (\boldsymbol{\nu}_i - \boldsymbol{\nu}_j) - 2 \ . \tag{5.13}$$

The hole diagram of any deep hole is an extended Coxeter–Dynkin diagram, and there are twenty-three possibilities for the corresponding root system, in exact correspondence with the twenty-three root systems of Niemeier lattices [36].[19] Any two holes with the same hole diagram can be related by automorphisms of the Leech lattice, and hence under $O(\Lambda_L)$ there are exactly twenty-three orbits of deep holes close to the origin. Finally, for any a deep hole $\mathfrak{m}$ with diagram having Coxeter number $h$ the vector $h\mathfrak{m}$ is a primitive element in $\Lambda_L$.[20]

### Leech deep holes and shift orbifolds

Consider a shift orbifold of the Leech theory, where we take $s$ to be a deep hole of type $\mathfrak{g}$, i.e. whose hole diagram is the extended Coxeter–Dynkin diagram of $\mathfrak{g}$ and has Coxeter number $h$. Without loss of generality we assume that $s$ is in the Voronoi cell, and hence that $s \cdot s = 2$. The vertices of the deep hole are located at a distance $(\boldsymbol{\nu}_i - s) \cdot (\boldsymbol{\nu}_i - s) = 2$ from $s$ and therefore satisfy

$$s \cdot \boldsymbol{\nu}_i = \tfrac{1}{2}\boldsymbol{\nu}_i \cdot \boldsymbol{\nu}_i \ . \tag{5.14}$$

Since $h\,s \in \Lambda_L$, we can perform a $\mathbb{Z}_h$ shift orbifold of the Leech CFT using $s$, leading to

$$(\Lambda_L)_{\text{inv}} = \{\, \boldsymbol{L} \in \Lambda_L \mid \boldsymbol{L} \cdot s \in \mathbb{Z} \,\} + \text{Span}_{\mathbb{Z}}\{s\} \ . \tag{5.15}$$

The invariant lattice contains the roots $\boldsymbol{\alpha}_i = -\boldsymbol{\nu}_i + s$ that span a root system isomorphic to $\mathfrak{g}$, and since the invariant lattice is even self-dual, it must be isomorphic to the Niemeier lattice

$$(\Lambda_L)_{\text{inv}} \simeq \Lambda_{\mathfrak{g}} \ . \tag{5.16}$$

Hence, any shift orbifold of the Leech theory with $s$ a deep hole leads to one of the twenty-three other Niemeier CFTs $\Lambda_{\mathfrak{g}}$, with $\mathfrak{g}$ given by the hole diagram of $s$. This shift symmetry is in fact the quantum symmetry of the Borcherds construction reviewed above, where the Leech theory is recovered as a specific shift orbifold of the $\Lambda_{\mathfrak{g}}$ theory.

### A characterization of Leech shift orbifolds

We can now obtain a complete picture of shift orbifolds of the Leech CFT. Consider a cyclic $\mathbb{Z}_k$ orbifold with shift vector $s$. Which of the Niemeier CFTs describes the orbifold theory?

---

[19]The extended Coxeter-Dynkin diagram associated to $\mathfrak{g}$ is obtained by replacing each simple factor in $\mathfrak{g}$ by the corresponding extended diagram.

[20]To see this, consider the root lattice spanned by the vectors $\boldsymbol{\alpha}_i = \mathfrak{m} - \boldsymbol{\nu}_i$. It is isomorphic to $\mathfrak{g}$, and for each simple component (with nodes labelled by a collection of integers $I$) there is a linear relation $\sum_{i \in I} \kappa_i \boldsymbol{\alpha}_i = 0$, where the Kac marks $\{\kappa_i\}_{i \in I}$ sum to $h$. Consequently, the deep hole satisfies $h\mathfrak{m} = \sum_{i \in I} \kappa_i \boldsymbol{\nu}_i$.

The answer is obtained using an algorithmic method, similar to the one described in section 4. We can take $s$ to be in the Voronoi cell, since for any vector $\boldsymbol{L}$ of the Leech lattice, $s$ and $s + \boldsymbol{L}$ describe the same orbifold. In particular, the shift vector satisfies

$$s \cdot s \leq 2 \ . \tag{5.17}$$

We fix an embedding of the Leech lattice in the Lorentzian lattice

$$\Gamma = \Lambda_{1,1} + \Lambda_{\mathrm{L}} \ , \tag{5.18}$$

such that any $\boldsymbol{p} \in \Gamma$ is uniquely written as $\boldsymbol{p} = \mathrm{w}\boldsymbol{e} + \mathrm{n}\boldsymbol{e}^* + \boldsymbol{L}$, where $\mathrm{w}, \mathrm{n}$ are integers and $\boldsymbol{L} \in \Lambda_{\mathrm{L}}$. The Leech lattice can be obtained by considering the sublattice $\boldsymbol{e}^{\perp} \subset \Gamma$ of points $\boldsymbol{p}$ orthogonal to $\boldsymbol{e}$ (those with $\mathrm{n} = 0$), and modding out this sublattice by $\boldsymbol{e}$ (which allows to also set $\mathrm{w} = 0$). Hence, we trivially have the isomorphism

$$\Lambda_{\mathrm{L}} \simeq \boldsymbol{e}^{\perp}/\boldsymbol{e} \ . \tag{5.19}$$

Similarly, we can describe the invariant lattice $(\Lambda_{\mathrm{L}})_{\mathrm{inv}}$ as a sublattice of $\Gamma$. To each lattice point of the orbifold theory $\widetilde{\boldsymbol{L}} = \boldsymbol{L} + \ell s$, with $\ell \in \{0, 1, \ldots, k-1\}$, we associate the element $\boldsymbol{p}_{\widetilde{\boldsymbol{L}}} \in \Gamma$ defined as

$$\boldsymbol{p}_{\widetilde{\boldsymbol{L}}} = \ell\boldsymbol{e} + (s \cdot \boldsymbol{L} + \tfrac{1}{2}\ell s \cdot s)\boldsymbol{e}^* + \boldsymbol{L} \ , \tag{5.20}$$

which has norm $\boldsymbol{p}_{\widetilde{\boldsymbol{L}}} \cdot \boldsymbol{p}_{\widetilde{\boldsymbol{L}}} = \widetilde{\boldsymbol{L}} \cdot \widetilde{\boldsymbol{L}}$. Such a point is orthogonal to the primitive null vector

$$\boldsymbol{f} = k \left( \boldsymbol{e} - \tfrac{s \cdot s}{2}\boldsymbol{e}^* - s \right) \ . \tag{5.21}$$

Moreover, any vector $\boldsymbol{p} \in \Gamma$ orthogonal to $\boldsymbol{f}$ can be uniquely written as $\boldsymbol{p} = \boldsymbol{p}_{\widetilde{\boldsymbol{L}}} + \widetilde{\mathrm{w}}\boldsymbol{f}$, for some integer $\widetilde{\mathrm{w}}$ and some $\widetilde{\boldsymbol{L}} \in (\Lambda_{\mathrm{L}})_{\mathrm{inv}}$. Hence, the invariant lattice is isomorphic to

$$(\Lambda_{\mathrm{L}})_{\mathrm{inv}} \simeq \boldsymbol{f}^{\perp}/\boldsymbol{f} \ . \tag{5.22}$$

We can proceed as in section 4. Since the shift vector $s$ is in the Voronoi cell and is subject to the orbifold consistency condition $s \cdot s \in \frac{2}{k}\mathbb{Z}$, its norm can be written as

$$\frac{s \cdot s}{2} = \frac{k'}{k} \ , \tag{5.23}$$

for some $k' \leq k$, with $k' = k$ if and only if $s$ is a deep hole. The Lorentzian lattice $\Gamma$ has automorphisms that are the analogues of the heterotic Wilson line shifts (4.9) and factorized duality (4.10), and for each vector of the Leech lattice $\boldsymbol{q}$ we define an automorphism $\varphi_{\boldsymbol{q}} \in \mathrm{O}(\Gamma)$

via

$$\varphi_{\boldsymbol{q}}(\mathrm{w}\boldsymbol{e} + \mathrm{n}\boldsymbol{e}^* + \boldsymbol{L}) = -\mathrm{n}\boldsymbol{e} - (\mathrm{w} - \boldsymbol{q} \cdot \boldsymbol{L} - \tfrac{1}{2}\boldsymbol{q} \cdot \boldsymbol{q}\mathrm{n})\boldsymbol{e}^* + \boldsymbol{L} + \boldsymbol{q}\mathrm{n} \ . \tag{5.24}$$

This duality maps the null vector $\boldsymbol{f}$ to $\boldsymbol{f}' = \varphi_{\boldsymbol{q}}(\boldsymbol{f})$, which we rewrite as

$$\boldsymbol{f}' = k'\left(\boldsymbol{e} - \tfrac{s'\cdot s'}{2}\boldsymbol{e}^* - s'\right) \ , \qquad\qquad \text{with} \qquad s' = \frac{k}{k'}s + \boldsymbol{q} \ . \tag{5.25}$$

We can choose $\boldsymbol{q}$ such that $s'$ is in the Voronoi cell of the Leech lattice. By construction, the invariant lattice $(\Lambda_{\mathrm{L}})_{\mathrm{inv}}$ is isomorphic to $\boldsymbol{f}'^{\perp}/\boldsymbol{f}'$. Hence we obtain an equivalent description of the orbifold theory, as a $\mathbb{Z}_{k'}$ orbifold with shift vector $s'$. This is exactly the dual shift orbifold described in section 3.4.

Repeating this procedure, we obtain a sequence of null vectors $\boldsymbol{f}, \boldsymbol{f}', \boldsymbol{f}'', \ldots$, and accordingly a sequence of shift vectors $s, s', s'', \ldots$, that lead to isomorphic orbifold theories. After a finite number of steps, the shift vectors converge to a point that is either the origin, or one of the deep holes of the Leech lattice, of type $\mathfrak{g}$. In the first case, the orbifold theory is the Leech CFT. In the other cases, the invariant lattice is isomorphic to the Niemeier lattice $\Lambda_{\mathfrak{g}}$. We note in passing that, since the integers $k \geq k' \geq k'' \geq \dots$ decrease at each step and reach a final value $h$ (the Coxeter number of the invariant lattice), we recover the Chenevier–Lannes bound $k \geq h$.

Consequently, we can unambiguously characterize all possible shift orbifolds of the Leech CFT, by applying the above algorithm to the corresponding shift vectors. The convergence of this algorithm, just like in the heterotic setting, relies on the fact that the Voronoi cell of the Leech lattice fits in a sphere of radius $\sqrt{2}$. This is not the case for the other Niemeier lattices, and it is unclear how to extend our results to those remaining $c = 24$ theories. Compared to the rank 16 case, the underlying difficulties can be traced back to the automorphism group of the Lorentzian lattice $\Gamma_{1,25}$, described in [37], which is considerably more intricate than the T-duality group $\mathrm{O}(\Gamma_{1,17})$.

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
