# Peer review of "Shift orbifolds, decompactification limits, and lattices"

_SciPost Physics_

## Round 1 · Referee Report · Anonymous (Referee 1) · 2025-11-15

Strengths

  • clear general discussion of a class of heterotic orbifolds,
  • poses an interesting problem concerning relation between even-self dual lattices and solves it in a insightful way connected to decompactification along an expectator circle.

Weaknesses

None

Report

In this article the authors consider shift orbifolds of a generic heterotic string in
$d$ dimensions. The latter, dubbed Narain CFT, is characterized by an even self-dual
lattice of signature $(d,16+d)$ and the shifts characterizing the action of the orbifold group are given in terms of lattice vectors. The construction is clearly explained in careful detail.

Although aspects of shift orbifolds have already been studied in the literature, this article
asks interesting questions and answers them in clever ways that make contact with decompactification
limits. In particular, the authors determine all shift orbifolds of order $k \le 5$ that exchange
the two heterotic lattices in 10 dimensions (the case $d=0$).

The authors also analyze shift orbifolds involving $c=24$ CFTs based on even self-dual lattices.
In particular, starting with the Leech lattice they find the corresponding Niemeier lattice upon
orbifolding.

The findings of this paper are of current interest as they have potential applications in the construction
of novel string compactifications. Moreover, they are neatly presented.
Publication in Scipost is recommended.

Requested changes

None

Recommendation

Publish (easily meets expectations and criteria for this Journal; among top 50%)

---

## Round 1 · Referee Report · Anonymous (Referee 2) · 2025-11-17

Strengths

1- The article is exceptionally clear and careful about all technical points, many of which are usually not explicitly written out in the literature.

2- The article presents in a very explicit way various useful technical connections between lattice theory and conformal field theory.

3- The article contains a thorough set of references including modern work on the topic, making it a particularly useful working reference.

Weaknesses

1- The main weakness of the article is that it does not present significantly novel results, but this is clearly acknowledged by the authors.

Report

The article presents a very thorough and detailed discussion of a topic that is seldom discussed explicitly in the literature, namely the role of shifts in orbifolds of string worldsheet CFTs. Given the current interest in asymmetric orbifolds of heterotic and type II strings, specially in classification efforts, I believe that this article will serve as an useful and reliable reference in this topic.

Recommendation

Publish (easily meets expectations and criteria for this Journal; among top 50%)

---

## Round 1 · Referee Report · Anonymous (Referee 3) · 2025-12-12

Report

The article studies orbifolds of a family of 2-dimensional conformal field theories known as Narain CFTs which describe string worldsheet theories. The type of the orbifold is shift orbifold and its generator is an element of a finite symmetry group of the Narain CFT. The orbifold theory is again a Narain CFT located at a different point in the Narain moduli space.

The authors focus on 10d heterotic string theory. They study shift orbifolds systematically and classify shifts vectors that exchange $E_8+E_8$ and Spin(32)$/\mathbb Z_2$ heterotic theories. To do so, they compactify the theory on a circle, define the shift orbifold of the Narain CFT including its action on the Narain lattice with signature $(17,1)$, and finally take two decompactification limits as the radius of the circle goes to $\infty$ and $0$ to recover the 10d theory.

They develop an algorithm to classify shift vectors that yield inequivalent lattices for abelian orbifold groups. They present the results for the first few low lying values of the order of the cyclic group for the 10d heterotic string theory. Furthermore, they apply their algorithm to the Leech lattice CFT by embedding the Leech lattice in the Narain lattice with signature $(25,1)$. For cyclic groups, they classify shift vectors and determine which of the 24 Niemeier CFTs (including the original Leech CFT) is obtained from the shift orbifold.

Although, as the authors have pointed out, the lattice shifting method and shift orbifold conformal field theories are well-known subjects and have been studied in different contexts, the novel aspect of the work lies in developing a general algorithm to construct conformal field theories with inequivalent lattices and to classify shift orbifolds. The paper presents interesting results and the algorithm can be extended to other families of conformal field theories and to other types of orbifolds which include shift vectors and therefore, will be useful in wider context. The paper is well-written and the lattice construction and choices of the shift vectors are explained clearly. We recommend the paper for publication in SciPost after minor revisions based on the following points.

1 ) Could the authors clarify why the phase the vertex operator acquires under the orbifold action is of the form given in equation (3.2)? How does it change if the lattice was even but not self-dual?

2) Could the authors specify the rank and signature of the invariant lattices in both heterotic and Niemeier theories.

3) Could the authors comment if and how the algorithm can be used in compactifications to lower dimensions with Narain lattices of signature $(16+d,d)$ and decompactifications to dimensions equal or less than 10? What are the obstructions?

4) How can the authors guarantee that the degeneracy of the states of the constructed shift orbifolds are non-negative integers in all untwisted and twisted sectors?

Recommendation

Publish (easily meets expectations and criteria for this Journal; among top 50%)

---

## Round 2 · Author Response

In response to the referees’ suggestions, we have made a few minor changes to clarify several points.

---

## Round 2 · List of Changes

• Footnote on page 6 explaining the form of the cocycle phase.
  • Clarifications on the twisted Hilbert space construction, at the beginning of subsection 3.1.
  • Explicit mention of the rank and signature of the orbifold lattice, below equation (3.20).
  • Comments on generalizations to heterotic compactifications in lower dimensions, at the end of subsection 4.3.

---

## Editorial Decision

in_refereeing